# A Unified Framework for Generalization Error Analysis of Learning with Arbitrary Discrete Weak Features

**Kosuke Sugiyama** [1]   **Masato Uchida** [1]

## Abstract

In many real-world applications, predictive tasks inevitably involve low-quality input features (Weak Features; WFs) which arise due to factors such as misobservations, missingness, or partial observations. While several methods have been proposed to estimate the true values of specific types of WFs and to solve a downstream task, a unified theoretical framework that comprehensively addresses these methods remains underdeveloped. In this paper, we propose a unified framework called Weak Features Learning (WFL), which accommodates arbitrary discrete WFs and a broad range of learning algorithms, and we demonstrate its validity. Furthermore, we introduce a class of algorithms that learn both the estimation model for WFs and the predictive model for a downstream task and perform a generalization error analysis under finite-sample conditions. Our results elucidate the interdependencies between the estimation errors of WFs and the prediction error of a downstream task, as well as the theoretical conditions necessary for the learning approach to achieve consistency. This work establishes a unified theoretical foundation, providing generalization error analysis and performance guarantees, even in scenarios where WFs manifest in diverse forms.

## 1. Introduction

The performance and explainability of machine learning models are highly dependent on the quality of the training data. However, in practical applications, obtaining high-quality data is often infeasible due to constraints such as

[1]Major in Computer Science and Communications Engineering, Waseda University, Tokyo, Japan. Correspondence to: Kosuke Sugiyama <kohsuke0322@asagi.waseda.jp>, Masato Uchida <m.uchida@waseda.jp>.

*Proceedings of the $42^{nd}$ International Conference on Machine Learning*, Vancouver, Canada. PMLR 267, 2025. Copyright 2025 by the author(s).

privacy concerns, high observation costs, and uncertainties in sensor measurements. Consequently, challenges often stem from low-quality labels (referred to as Weak Labels; WLs) that contain incorrect, incomplete, or ambiguous supervisory information, and low-quality features (referred to as Weak Features; WFs), which manifest as misobservations, missing values, or ambiguous observations. Extensive research has focused on WLs within the framework of weakly supervised learning (WSL), covering methods such as semi-supervised learning and learning with label noise, which have provided substantial theoretical guarantees (Chapelle et al., 2006; Elkan & Noto, 2008; Cour et al., 2011; Natarajan et al., 2013; Ishida et al., 2017). In contrast, research on WFs has proposed methods such as impute-then-regress (ItR), which imputes missing before regression (Josse et al., 2024; Bertsimas et al., 2021; Le Morvan et al., 2020b; 2021), and complementary features learning (CFL), which leverages complementary features (CFs) that differ from the true values (Sugiyama & Uchida, 2024). However, a unified framework that provides consistent theoretical guarantees across the various forms of WFs remains underexplored.

Motivated by these challenges, this study focuses on weak features learning (WFL), a generalized learning problem involving arbitrary WFs. A common approach involves sequentially or iteratively learning *feature estimation models $g$* to estimate the true values of WFs (referred to as exact values) and a *label prediction model $f$* to predict downstream task labels using the outputs of $g$. This strategy is considered rational for improving both the quality of WFs and the predictive performance of a downstream task (Yoon et al., 2018; Mattei & Frellsen, 2019; Le Morvan et al., 2020a; Ipsen et al., 2021; Zaffran et al., 2023; Ipsen et al., 2022; Sugiyama & Uchida, 2024). Indeed, prior approaches such as ItR and CFL are grounded in this learning strategy, utilizing various machine learning methods to construct $g$. These methods aim to improve the quality of WFs, thereby enhancing the generalization performance and explainability of $f$ in a downstream task.

However, in the context of WFL, where various methods have been proposed, significant gaps remain in understanding how the learning of $g$ and $f$ impacts each other's learning efficiency and under what conditions WFL can achieve

optimal $g$ and $f$ (i.e., consistency). For example, in ItR, while the conditions under which $g$ and $f$ become Bayes rules have been analyzed, the generalization error analysis under finite samples and any data distribution has not yet been sufficiently conducted (Josse et al., 2024; Bertsimas et al., 2021; Le Morvan et al., 2020b; 2021). Also, in CFL, which addresses CFs, theoretical analysis remains underdeveloped, and clear guidelines for handling diverse forms of WFs in a unified manner have yet to be established (Sugiyama & Uchida, 2024).

This study aims to investigate the mutual influence between estimating the exact values of WFs and learning a downstream task. Specifically, we focus on scenarios involving discrete WFs (hereafter referred to as discrete WFL) and provide a unified formalization. By performing finite-sample error analysis for a generalized class of learning algorithms, we systematically address these questions. The proposed theoretical framework extends beyond situations involving missing values or CFs. It also accommodates cases where WFs arise in the diverse forms discussed in the WLs literature, such as erroneous observations or scenarios where only a candidate set containing the exact value is observed (Natarajan et al., 2013; Cour et al., 2011; Feng et al., 2020; Xu et al., 2021). Consequently, the framework not only reinterprets existing approaches such as ItR and CFL but also offers new theoretical insights into previously unexplored WF settings. Furthermore, as the class of learning algorithms analyzed in this study encompasses various existing methods (Yoon et al., 2018; Mattei & Frellsen, 2019; Ipsen et al., 2021; Josse et al., 2024; Le Morvan et al., 2021; Ipsen et al., 2022; Sugiyama & Uchida, 2024), our results provide a unified theoretical evaluation framework for these methods. We have also developed a unified formulation and analysis for scenarios involving continuous WFs, and the results for continuous WFs are presented in a separate paper (Sugiyama & Uchida, 2025).

The main contributions of this study are as follows:

1. We propose a risk-based formulation to address arbitrary discrete WFs and demonstrate that the introduced objective function facilitates the learning of $f$, which captures the true input-output relationship. This validates the proposed formulation (Section 3.2).

2. Within the proposed formulation, we define the Learning Algorithm Class for discrete WFL (LAC-dWFL), which flexibly combines three steps: (i) learning $g$ using WFs as weak supervision, (ii) learning $f$ with a fixed $g$, and (iii) learning $g$ with a fixed $f$. This framework accommodates both sequential and iterative learning approaches, offering a unified perspective on diverse methods (Section 3.3).

3. For step (ii), we derive the error bound for $f$ given any fixed $g$ (Section 4.2), providing theoretical insights into how

the estimation errors of $g$ influence the error bound for $f$. By integrating the theoretical framework of WSL in (i), we further analyze how the properties of WFs influence $f$'s error bound, how the order of the error bound evolves with the learning of $g$, and the conditions under which sequential learning in LAC-dWFL (performing steps (i) and (ii) in sequence) achieves consistency.

4.For step (iii), we evaluate the generalization error of $g$ given any fixed $f$ (Section 4.3) and analyze how the properties and generalization performance of $f$ affect the error bound for $g$. By integrating the results of Contributions 3 and 4, we establish the conditions under which iterative learning in LAC-dWFL (alternating steps (ii) and (iii)) achieves the consistency for $f \circ g$.

5. We validate the proposed theoretical framework on real-world datasets to evaluate how accurately the derived error bounds reflect actual learning behaviors (Section 5).

## 2. Related work

### 2.1. Learning Problems involving WFs

ItR and CFL are representative learning problems that serve as special cases of WFL, involving specific types of WFs. This section provides an overview of these frameworks.

In ItR, the focus lies on scenarios where input features contain missing values. The fundamental approach of ItR involves imputing the missing values and utilizing the resulting complete dataset to learn a label prediction model $f$ for a downstream task. Sequential learning methods have been proposed, employing feature estimation models $g$ that utilize techniques such as constant imputation (Josse et al., 2024) or machine learning-based approaches (Yoon et al., 2018; Mattei & Frellsen, 2019; Ipsen et al., 2021). Furthermore, joint or iterative optimization of $g$ and $f$ has also been explored in the literature (Le Morvan et al., 2020a; Ipsen et al., 2022).

Theoretical analyses of ItR have investigated whether combinations of $g$ and $f$ exist that equal Bayes rule, focusing on regression or classification as downstream tasks (Josse et al., 2024; Bertsimas et al., 2021; Le Morvan et al., 2021). However, these studies do not address the existence of learning algorithms capable of constructing optimal $g$ and $f$. Moreover, analyses in finite-sample settings have been confined to restrictive cases where the true $f$ is assumed to be a linear model, leaving more general problem settings unexplored (Le Morvan et al., 2020b). Additionally, these analyses often restrict downstream tasks to regression or classification and, in some cases, confine the loss function of $f$ to mean squared error (Le Morvan et al., 2020b; 2021). In this paper, while we focus on discrete WFs, we achieve a generalization error analysis of WFL in finite-sample settings by assum-

ing only the boundedness and Lipschitz continuity of the loss function of $f$, without imposing any constraints on a downstream task and a data distribution.

In CFL, the primary focus lies on learning problems where input features include CFs (Sugiyama & Uchida, 2024). Within this framework, the feature estimation model $g$ and the label prediction model $f$ are defined as probabilistic models, and an objective function leveraging the Kullback-Leibler divergence has been derived to learn $g$ and $f$. For learning $g$ in CFL, complementary label learning (CLL) (Ishida et al., 2017; 2019; Yu et al., 2018; Lin & Lin, 2023; Ruan et al., 2024), a weakly supervised learning approach that predicts true labels from datasets labeled exclusively with incorrect labels, can be employed. Moreover, since partial label learning (PLL) (Cour et al., 2011; Feng et al., 2020; Xu et al., 2021; Tian et al., 2023), where supervision is provided in the form of a set containing the true label, can be interpreted as a generalized learning problem of CLL (Katsura & Uchida, 2020). Therefore, PLL is also applicable to the learning of $g$.

However, in CFL, the learning behavior under finite samples, as well as the conditions required to obtain asymptotically optimal $g$ and $f$, have yet to be theoretically clarified. In this paper, we restrict $g$ and $f$ to deterministic models and perform a generalization error analysis of WFL for arbitrary downstream task and bounded, Lipschitz-continuous loss functions, thereby shedding light on the theoretical properties of CFL.

### 2.2. WFs Whose Exact Values Can Be Estimated

When constructing a feature estimation model $g$, it is natural to treat the observed values of each WF as WLs and employ the WSL methods. In fact, depending on the types and settings of WFs, $g$ can be learned using WSL methods. In WSL, a variety of WL settings have been studied, including the aforementioned CLL and PLL. For instance, noisy label learning (Natarajan et al., 2013) deals with learning from data containing incorrect labels, while positive-unlabeled learning (Elkan & Noto, 2008) focuses on learning binary classifiers using only positive and unlabeled samples.

Various WSL methods have been theoretically formulated, and their generalization error has been analyzed under finite-sample conditions (Cour et al., 2011; Feng et al., 2020; Xu et al., 2021; Natarajan et al., 2013; Ishida et al., 2017; Yu et al., 2018). Many of these objective functions were defined using unbiased estimators of the expected risk in supervised learning, the expected risk's upper bounds that are computable with WLs, or risks whose optimal solutions align those of the expected risk. Their theoretical analyses elucidated the conditions under which optimal hypotheses can be obtained by minimizing each objective function, as well as the relationship between WL settings and the error

bounds. Thus, if WSL is employed to learn $g$ in WFL, the learning of $g$ alone can be analyzed based on the WSL theories.

However, in WFL, it is necessary to consider the learning of both $g$ and $f$. For example, in sequential learning, the learning of $f$ depends on the output of $g$. In iterative learning, the learning of $g$ depends on $f$, unlike the case where $g$ is learned solely using WSL. Therefore, a theoretical discussion that establishes the relationship between $g$ and $f$ is essential. In this paper, we perform a generalization error analysis of WFL, explicitly considering the relationship between $g$ and $f$.

## 3. Formulation

### 3.1. Review of ERM

In this paper, we formulate WFL from the perspective of risk minimization and adopt empirical risk minimization (ERM) as the learning framework. Below, we briefly review ERM in ordinary supervised learning (Shalev-Shwartz & Ben-David, 2014; Mohri et al., 2018). Let the input space be $\mathcal{X} \subseteq \mathbb{R}^d$ and the label space be $\mathcal{Y} \subseteq \mathbb{R}$. Here, $d \in \mathbb{N}_+$ represents an input dimension. We denote the random variables representing an instance by $\boldsymbol{X}$ and the random variable representing labels by $Y$, assuming that $(\boldsymbol{X}, Y)$ follows the true distribution $p_*(\boldsymbol{x}, y)$ over $\mathcal{X} \times \mathcal{Y}$ independently and identically distributed (i.i.d.). The goal in the ERM framework is to find a label prediction model $f : \mathcal{X} \to \mathbb{R} \in \mathcal{F}$ that minimizes the expected risk:

$$R_l(f) := \mathbb{E}_{p_*(\boldsymbol{x}, y)}[l(f(\boldsymbol{X}), Y)], \qquad (3.1)$$

where $l : \mathbb{R} \times \mathbb{R} \to \mathbb{R}_+$ is a loss function, and $\mathcal{F}$ is the hypothesis set of label prediction models. Since only finite samples are available in practice, ERM approximates the expected risk with the empirical risk computed as a sample average and learns $f$ by minimizing this empirical risk.

### 3.2. Formulation of discrete WFL

In this section, we formulate WFL based on risk minimization and employ ERM as the learning method. To account for the presence of WFs in instances, we decompose $\boldsymbol{X}$ representing an instance, into $\boldsymbol{X}^{\mathrm{w}}$ representing the exact values of WFs and $\boldsymbol{X}^{\mathrm{o}}$ representing the remaining ordinary features (OFs), such that $\boldsymbol{X} = (\boldsymbol{X}^{\mathrm{w}}, \boldsymbol{X}^{\mathrm{o}})$. Let $\mathcal{X}^{\mathrm{w}}$ and $\mathcal{X}^{\mathrm{o}}$ denote the domains of $\boldsymbol{X}^{\mathrm{w}}$ and $\boldsymbol{X}^{\mathrm{o}}$, respectively, with $\mathcal{X}^{\mathrm{w}} \times \mathcal{X}^{\mathrm{o}} = \mathcal{X}$. Here, $\mathcal{X}^{\mathrm{w}}$ is assumed to be a finite set. We denote the observed values of WFs as the random variables $\overline{\boldsymbol{X}}^{\mathrm{w}}$, which follows the probability distribution $\bar{p}_*(\bar{\boldsymbol{x}}^{\mathrm{w}}|\boldsymbol{x}, y)$.

We define the feature estimation models for estimating the exact values of WFs as $\boldsymbol{g} := (g_1, \ldots, g_{F^{\mathrm{w}}}) \in \mathcal{G} := \mathcal{G}_1 \times \cdots \times \mathcal{G}_{F^{\mathrm{w}}} : \mathcal{X}^{\mathrm{o}} \to \mathcal{X}^{\mathrm{w}}$, where $F^{\mathrm{w}}$ is the number of

WFs, and each $g_j \in \mathcal{G}_j : \mathcal{X}^{\mathrm{o}} \to \mathcal{X}_j^{\mathrm{w}}, \forall j \in [F^{\mathrm{w}}]$. Here, $[F^{\mathrm{w}}] := \{1, \ldots, F^{\mathrm{w}}\}$, and $\mathcal{G}_j$ represents the hypothesis set for estimating $X_j^{\mathrm{w}}$. The probability mass function (PMF) representation of $\boldsymbol{g}$ is defined as $q_{\boldsymbol{g}}(\boldsymbol{x}^{\mathrm{w}}|\boldsymbol{x}^{\mathrm{o}}) := \mathbb{1}_{[\boldsymbol{x}^{\mathrm{w}}=\boldsymbol{g}(\boldsymbol{x}^{\mathrm{o}})]}$. For simplicity, this paper primarily focuses on binary classification, but the proposed formulation and analyses can be easily extended to other prediction tasks such as multi-class classification or regression.

The primary objectives of WFL are to improve the generalization performance of a downstream task and to restore explainability lost due to WFs. The primary factor reducing explainability is the inaccuracy of information provided by WFs . The most natural approach to address this issue is to estimate the exact values of WFs accurately. Accordingly, WFL aims to learn $f$ and $\boldsymbol{g}$ that minimize the following two risks. The first risk evaluates the generalization error of $f$:

$$
\begin{aligned}
R_{l,\boldsymbol{g}}(f) &:= \mathbb{E}_{p_*(\boldsymbol{x}^{\mathrm{o}}, y) q_{\boldsymbol{g}}(\boldsymbol{x}^{\mathrm{w}}|\boldsymbol{x}^{\mathrm{o}})}[l(f(\boldsymbol{X}), Y)] \\
&= \mathbb{E}_{p_*(\boldsymbol{x}^{\mathrm{o}}, y)}[l(f(\boldsymbol{g}(\boldsymbol{X}^{\mathrm{o}}), \boldsymbol{X}^{\mathrm{o}}), Y)].
\end{aligned} \tag{3.2}
$$

The second risk that assesses the estimation errors of $\boldsymbol{g}$:

$$
R_{01,j}(g_j) := \mathbb{E}_{p_*(\boldsymbol{x})}[l_{01}(g_j(\boldsymbol{X}^{\mathrm{o}}), X_j^{\mathrm{w}})], \forall j \in [F^{\mathrm{w}}]. \tag{3.3}
$$

Here, $l_{01}(y, y') := \mathbb{1}_{[y \neq y']}$ represents the 0-1 loss. Finally, the objective function for discrete WFL is defined as a linear combination of these risks:

$$
R_{l,\lambda}^{\mathrm{dWFL}}(\boldsymbol{g}, f) := R_{l,\boldsymbol{g}}(f) + \lambda \sum_{j \in [F^{\mathrm{w}}]} R_{01,j}(g_j), \tag{3.4}
$$

where $\lambda \in \mathbb{R}_+$ is a weighting parameter.

The objective function $R_{l,\lambda}^{\mathrm{dWFL}}$ facilitates the unified treatment of any discrete WFs. This unification arises from representing the error of $g_j$ via the risk $R_{01,j}$, which is aimed to be minimized regardless of the type of WF. In practice, for various types of WFs, WSL methods that learn $g_j$ aim to minimize $R_{01,j}$, by minimizing risks that serve as upper-bound of $R_{01,j}$ or risks whose optimal solutions align with those of $R_{01,j}$ (Cour et al., 2011; Feng et al., 2020; Natarajan et al., 2013; Ishida et al., 2017; Yu et al., 2018). Therefore, such WSL methods can be utilized to learn $g_j$ as part of minimizing $R_{l,\lambda}^{\mathrm{dWFL}}$.

The validity of our formulation is demonstrated by the following theorem [1]. Its proof is given in Appendix A.1.

**Theorem 3.1.** *For any $f \in \mathcal{F}$, $\boldsymbol{g} \in \mathcal{G}$, and $l$ bounded by $U_l < \infty$, the following inequality holds:*

$$
R_l(f) \leq R_{l,\boldsymbol{g}}(f) + U_l \sum_{j \in [F^{\mathrm{w}}]} R_{01,j}(g_j). \tag{3.5}
$$

---

[1] Theorem 3.1 can also be derived from Lemma 4.1, which is introduced later in Section 4.1. However, while Lemma 4.1 is derived for the theoretical analysis in Section 4, Theorem 3.1 is intended to establish the validity of our formalization. We introduce this theorem to enhance the clarity of our discussion.

The RHS of Eq.(3.5) equals $R_{l,U_l}^{\mathrm{dWFL}}$. By scaling $l$, $U_l$ can be aligned with any $\lambda$, and minimizing $R_{l,\lambda}^{\mathrm{dWFL}}$ is expected to yield an $f$ that also minimizes $R_l$. In other words, by using $R_{l,\lambda}^{\mathrm{dWFL}}$, $f$ is learned to capture the true relationship between $\boldsymbol{X}$ and $Y$, despite relying on $\overline{\boldsymbol{X}}^{\mathrm{w}}$.

This result of Theorem 3.1 is directly applicable to the following two scenarios. The first scenario occurs when test instances contain the exact values of WFs. For instance, practical scenarios arises when WFs are observed during training, but exact values are available during testing. In this scenario, minimizing $R_l$ is essential, but it cannot be computed directly from the training data containing WFs. In contrast, since Theorem 3.1 ensures that minimizing $R_{l,\lambda}^{\mathrm{dWFL}}$ indirectly minimizes $R_l$, our framework is a valuable approach for overcoming incomplete inputs during training while enhancing performance with complete inputs at test time.

The second scenario involves training data containing a mix of instances with exact values of WFs $\boldsymbol{X}^{\mathrm{w}}$ and instances with WFs $\overline{\boldsymbol{X}}^{\mathrm{w}}$. In practical applications, some portions of the data may be observed in detail, yielding exact values, while other portions may be only partially observed. Theorem 3.1 guarantees that minimizing $R_{l,\lambda}^{\mathrm{dWFL}}$ contributes to minimizing $R_l$. Thus, it enables the simultaneous use of both data types during training. This approach facilitates the development of learning methods that seamlessly integrate both data types into a unified framework.

### 3.3. Learning Algorithm Class for discrete WFL

In this section, we introduce a class of learning algorithms that uniformly address not only any discrete WFs but also diverse methods within the discrete WFL framework. Building on the formulation proposed in Section 3.2, we define a class that encompasses numerous existing methods in ItR and CFL as follows:

**Definition 3.2** (LAC-dWFL). *learning algorithm class for discrete WFL (LAC-dWFL) refers to the set of algorithms in discrete WFL that learn the feature estimation models $\boldsymbol{g}$ and the label prediction model $f$ using one or a combination of the following three steps:*

(i) Learning $\boldsymbol{g}$ by using $\overline{\boldsymbol{X}}^{\mathrm{w}}$ as weak supervision and minimizing $\sum_{j \in [F^{\mathrm{w}}]} R_{01,j}$, either directly or indirectly.

(ii) Learning $f$ with $\boldsymbol{g}$ fixed by minimizing $R_{l,\boldsymbol{g}}$.

(iii) Learning $\boldsymbol{g}$ with $f$ fixed by minimizing $R_{l,\lambda}^{\mathrm{dWFL}}$.

The introduction of LAC-dWFL allows for a unified treatment of a wide range of methods for WFL. Most methods applicable to ItR and CFL fall under the category of *sequential learning*, where steps (i) and (ii) are executed in sequence

(Yoon et al., 2018; Mattei & Frellsen, 2019; Ipsen et al., 2021; Josse et al., 2024; Le Morvan et al., 2021; Sugiyama & Uchida, 2024). Additionally, methods that represent $\boldsymbol{g}$ and $f$ as neural networks and combine them (Le Morvan et al., 2020a; Ipsen et al., 2022) can be regarded as *iterative learning*, where steps (ii) and (iii) are executed repeatedly, when these components are alternately optimized. Such methods are thus encompassed within LAC-dWFL.

# 4. Theoretical analysis

This section presents a theoretical analysis of the unified learning algorithm class, LAC-dWFL. Through this analysis, we elucidate the common properties of LAC-dWFL and establish a foundation for the theoretical exploration of various methods encompassed within this class. To achieve this, it is necessary to elucidate the properties of steps (i), (ii), and (iii) within LAC-dWFL. As established in Section 3.3, step (i) involves WSL methods, and its properties can therefore be analyzed using existing WSL theories. Consequently, our theoretical focus is directed toward steps (ii) and (iii). In Section 4.1, we derive a fundamental inequality to analyze steps (ii) and (iii). In Section 4.2, we establish an error bound for $f$ learned via step (ii), given any $\boldsymbol{g}$. In Section 4.3, we establish an error bound for $\boldsymbol{g}$ learned via step (iii), given any $f$.

## 4.1. Deriving an Analytical Tool

Our objective is to examine how the learning of $f$ in step (ii) and $\boldsymbol{g}$ in step (iii) depend on the performance of $\boldsymbol{g}$ and $f$, respectively. To achieve this, the error bounds for $f$ and $\boldsymbol{g}$ must be expressed in terms of $R_{01,j}$ for any $j \in [F^{\mathrm{w}}]$ and $R_l$, respectively. To achieve this requirement, we present the following lemma, with its proof provided in Appendix A.2.

**Lemma 4.1.** *For any measurable $l$ bounded by $U_l < \infty$, $f \in \mathcal{F}$ and $\boldsymbol{g} \in \mathcal{G}$, the following holds:*

$$|R_l(f) - R_{l,\boldsymbol{g}}(f)| \leq$$
$$\left(\sqrt{R_l(f)} + \sqrt{R_{l,\boldsymbol{g}}(f)}\right)\left(2U_l \sum_{j \in [F^{\mathrm{w}}]} R_{01,j}(g_j)\right)^{\frac{1}{2}}. \tag{4.6}$$

Equation (4.6) shows that $|R_l(f) - R_{l,\boldsymbol{g}}(f)| = 0$ is achieved when either $\sum_{j \in [F^{\mathrm{w}}]} R_{01,j}(g_j) = 0$ or $\sqrt{R_l(f)} + \sqrt{R_{l,\boldsymbol{g}}(f)} = 0$. Although intuitive, this inequality serves a critical role in deriving subsequent error bounds. Lemma 4.1 enables the analysis of how $f$ and $\boldsymbol{g}$ influence each other's learning processes.

## 4.2. Analysis of Learning Label Prediction Model $f$

In this section, we conduct a theoretical analysis of the learning process of $f$ under LAC-dWFL's step (ii), where $\boldsymbol{g}$

remains fixed. Since step (ii) involves learning $f$ based on the output of $\boldsymbol{g}$ rather than $\boldsymbol{X}^{\mathrm{w}}$, the relationship between $\boldsymbol{g}$ and $f$ cannot be analyzed using ordinary supervised learning frameworks (Mohri et al., 2018). We derive an error bound for $f$ that captures how the errors of $\boldsymbol{g}$ influence the learning process of $f$, enabling an analysis of the step (ii).

To facilitate the analysis, we introduce the following definitions. Given $n \in \mathbb{N}_+$ training samples, we define the *ordinary dataset*, $S = \{(\boldsymbol{x}_i, y_i)\}_{i=1}^n$, and the *weak dataset*, $\overline{S} = \{(\bar{\boldsymbol{x}}_i^{\mathrm{w}}, \boldsymbol{x}_i^{\mathrm{o}}, y_i)\}_{i=1}^n$. Here, $\boldsymbol{x}_i = (\boldsymbol{x}_i^{\mathrm{o}}, \boldsymbol{x}_i^{\mathrm{w}})$, $y_i$ and $\bar{\boldsymbol{x}}_i^{\mathrm{w}}$ represent the realizations of $\boldsymbol{X} = (\boldsymbol{X}^{\mathrm{w}}, \boldsymbol{X}^{\mathrm{o}})$, $Y$ and $\overline{\boldsymbol{X}}^{\mathrm{w}}$, respectively. Also, the $i$-th samples in $S$ and $\overline{S}$ correspond to the same instance, for any $i \in [n]$. We assume that $\{(\boldsymbol{x}_i, y_i, \bar{\boldsymbol{x}}_i^{\mathrm{w}})\}_{i=1}^n$ are independently drawn from $p_*(\boldsymbol{x}, y)\bar{p}_*(\bar{\boldsymbol{x}}^{\mathrm{w}}|\boldsymbol{x}, y)$. Let $\widehat{R}_l$ and $\widehat{R}_{l,\boldsymbol{g}}$ denote the empirical risks calculated by the sample average over $S$ and $\overline{S}$, respectively. For any $\boldsymbol{g} \in \mathcal{G}$, the empirical risk minimizer obtained from LAC-dWFL's step (ii) is defined as follows:

$$f_{\boldsymbol{g},\overline{S}} := \arg\min_{f \in \mathcal{F}} \widehat{R}_{l,\boldsymbol{g}}(f).$$

Using Lemma 4.1, we establish the error bound for $f_{\boldsymbol{g},\overline{S}}$ learned via LAC-dWFL' step (ii) in the following theorem. The proof is provided in Appendix A.3.

**Theorem 4.2.** *Let $S$ and $\overline{S}$ be the ordinary dataset and weak dataset, respectively, each containing $n$ samples. For any measurable $\boldsymbol{g} \in \mathcal{G}$, $L_l$-Lipschitz continuous $l$ bounded by $U_l < \infty$ and $\delta \in (0,1)$, the following holds with probability at least $1 - \delta$:*

$$R_{l,\boldsymbol{g}}(f_{\boldsymbol{g},\overline{S}}) - R_l(f_{\mathcal{F}}) \leq$$
$$4\left(L_l \mathfrak{R}_n^*(\mathcal{F}) + L_l \mathfrak{R}_n^{\boldsymbol{g}}(\mathcal{F}) + U_l \sqrt{\frac{\log(4/\delta)}{2n}}\right)$$
$$+ \left\{2\left(R_l(f_{\mathcal{F}}) + 4L_l \mathfrak{R}_n^*(\mathcal{F}) + 2U_l \sqrt{\frac{\log(4/\delta)}{2n}}\right)^{\frac{1}{2}} \right. \tag{4.7}$$
$$+ \left(2U_l \sum_{j \in [F^{\mathrm{w}}]} R_{01,j}(g_j)\right)^{\frac{1}{2}}\right\}$$
$$\times \left(2U_l \sum_{j \in [F^{\mathrm{w}}]} R_{01,j}(g_j)\right)^{\frac{1}{2}}.$$

*Here, $f_{\mathcal{F}} := \arg\min_{f \in \mathcal{F}} R_l(f)$ represents the true risk minimizer in ordinary supervised learning.*

The terms $\mathfrak{R}_n^*(\mathcal{F})$ and $\mathfrak{R}_n^{\boldsymbol{g}}(\mathcal{F})$ represent the Rademacher complexities of the hypothesis class $\mathcal{F}$ under the distributions $p_*(\boldsymbol{x})$ and $p_*(\boldsymbol{x}^{\mathrm{o}})q_{\boldsymbol{g}}(\boldsymbol{x}^{\mathrm{w}}|\boldsymbol{x}^{\mathrm{o}})$, respectively, and measure the complexity of $\mathcal{F}$. Equation (4.7) reveals that the errors of $\boldsymbol{g}$ combine with the result of virtual ordinary supervised learning $(R_l(f_{\mathcal{F}}) + \cdots)^{1/2}$ to affect the error bound for $f$ in WFL.

The first contribution of Theorem 4.2 is its ability to reveal the following property concerning the learning of $f$ under LAC-dWFL's step (ii). Theorem 4.2 demonstrates how the

convergence rate of the error bound for $f$ with respect to $n$ depends on the estimation errors of $g$. Given the established result that the order of the Rademacher complexity's upper bound is $\mathcal{O}_p(1/n^{1/2})$ for kernel ridge regression and multilayer perceptrons (Mohri et al., 2018; Neyshabur et al., 2015), we assume the orders of Rademacher complexities about $\mathcal{F}$ or $\mathcal{G}$ are $\mathcal{O}_p(1/n^{1/2})$. Additionally, assume that $\mathcal{F}$ is sufficiently expressive and that $R_l(f_{\mathcal{F}}) = 0$. Under these assumptions, and with $g$ fixed, the first and second terms in the error bound have orders of $\mathcal{O}_p(1/n^{1/2})$ and $\mathcal{O}_p(1/n^{1/4})$, respectively. Therefore, the second term, which decreases more slowly, becomes dominant when $\sum_{j \in [F^w]} R_{01,j}$ is large. This result suggests that when learning $g$ to improve the generalization performance of $f$, maximizing the estimation accuracy of $g$ is sufficient, rather than tailoring the outputs of $g$ to be specifically suited for $f$. Furthermore, when WFs contain ambiguous information such as CFs, their values $\overline{X}^w$ can serve as inputs to $f$. The choice between $\overline{X}^w$ and the learned $g$ is a critical decision, empirically validated in the context of CFL (Sugiyama & Uchida, 2024). Theorem 4.2 concludes that, if the learned $g$ estimates $X^w$ more accurately than $\overline{X}^w$, using $g$ contributes more significantly to reducing the error bound for $f$ compared to directly using $\overline{X}^w$.

The second contribution of Theorem 4.2 is its ability to integrate our WFL theory with the theories of WSL methods used to learn $g$ under LAC-dWFL's step (i). This is because, when $g_j$ is learned using certain WSL methods, their theoretical frameworks enable the derivation of error bounds for $R_{01,j}(g_j)$ in Eq. (4.7) (Cour et al., 2011; Feng et al., 2020; Xu et al., 2021; Natarajan et al., 2013; Ishida et al., 2017; Yu et al., 2018). For example, when $\overline{X}_j^w$ is a CF, applying the CLL method by Ishida et al. (Ishida et al., 2017) for learning $g_j$, and assuming that $\mathcal{G}_j$ is sufficiently large to satisfy $\min_{g_j \in \mathcal{G}_j} R_{01,j}(g_j) = 0$, the following holds for any $L_l$-Lipschitz continuous $l$ and any $\delta \in (0, 1)$ with probability at least $1 - \delta$:

$$R_{01,j}(\hat{g}_j) \leq 4|\mathcal{X}_j^w|(|\mathcal{X}_j^w| - 1)L_l\mathfrak{R}_n^*(\mathcal{G}_j) \\ + (|\mathcal{X}_j^w| - 1)\sqrt{\frac{8\log(2/\delta)}{n}}. \tag{4.8}$$

The combination of such error bound with Eq. (4.7) enables a unified generalization error analysis for the sequential learning of $g$ and $f$ under LAC-dWFL's steps (i) and (ii), elucidating the following three aspects: Firstly, the combination enables the analysis the influence of WFs' properties on the learning of $f$. For instance, since Eq. (4.8) depends on $|\mathcal{X}_j^w|$, combining it with our bound allows for analyzing how the number of possible values $|\mathcal{X}_j^w|$ influence $f$'s learning. Secondly, this combination elucidates the impact of whether $g$ is learned or not on the learning of $f$. Applying Eq. (4.8) to Eq. (4.7) demonstrates that the order of the error bound for $f$ is $\mathcal{O}_p(1/n^{1/2})$. Thus, when a constant

value or $\overline{X}^w$ is used as $g(X^o)$, the order of the error bound remains $\mathcal{O}_p(1/n^{1/4})$, whereas learning $g$ improves this order to $\mathcal{O}_p(1/n^{1/2})$. In addition, Theorem 4.2 theoretically connects the error bounds of $f$ and $g$, thereby elucidating the conditions under which sequential learning achieves consistency, as following theorem. The proof is shown in Appendix A.4.

**Theorem 4.3.** *Assume the existence of true deterministic functions $g_j^* : \mathcal{X}^o \to \mathcal{X}_j^w$ for all $j \in [F^w]$, such that $(g_1^*, \ldots, g_{F^w}^*) \in \mathcal{G}$, and $f^* : \mathcal{X} \to \mathcal{Y}$ such that $f^* \in \mathcal{F}$. Additionally, suppose $l$ bounded by $U_l < \infty$ is $L_l$-Lipschitz continuous, and $\mathfrak{R}_n^*(\mathcal{F})$ and $\mathfrak{R}_n^g(\mathcal{F})$ asymptotically approach $0$ as $n \to \infty$. If, for all $j \in [F^w]$, the number of samples available for learning $g_j$ tends to infinity as $n \to \infty$, and a consistent method is employed to learn $g_j$, then sequential learning achieves consistency (i.e., as $n \to \infty$, $R_{l,g}(f_{g,S}) \to R_l(f_{\mathcal{F}})$).*

Thus, under the conditions stated in Theorem 4.3, an asymptotically optimal pair of $g$ and $f$ can be obtained through sequential learning alone. In contrast, in practical finite-sample scenarios, iterative learning involving LAC-dWFL's steps (ii) and (iii) may be necessary (Le Morvan et al., 2020b; 2021). In Section 4.3, we examines the step (iii) for a comprehensive understanding of LAC-dWFL.

### 4.3. Analysis of Learning Feature Estimation Models $g$

This section provides a theoretical analysis of the learning of $g$ with $f$ fixed in LAC-dWFL's step (iii). We aims to reveal how the learning of $g$ using $R_{l,\lambda}$, with $f$ fixed, is influenced by $f$.

The learning of $g$ using $R_{l,\lambda}^{\text{dWFL}}$ in Eq. (3.4) involves the simultaneous minimization of two risks, which makes it challenging to conduct generalization error analysis for ERM directly. In contrast, in ordinary supervised learning, the simultaneous minimization of an expected risk and a regularization term is formulated as structural risk minimization (SRM), which has established methods for generalization error analysis (Mohri et al., 2018). SRM selects a hypothesis from a restricted hypothesis class in which the regularization term is below a certain threshold and derives error bounds for the selected hypothesis. Focusing on the influence of $f$ on the learning of $g$, we treat $R_{l,f}$ as the expected risk for which bounds are derived and $\sum_{j \in [F^w]} R_{01,j}$ as the regularization term, and apply SRM theory.

To apply SRM theory, for any $j \in [F^w]$, we introduce the following definitions. Let $l_j$ denote the loss function for $g_j$ computed using $\overline{X}_j^w$. Define the datasets $\overline{S}_j := \{(\bar{x}_{ij}^w, x_i^o)\}_{i=1}^n$. Let $\overline{R}_{l_j}(g_j) := \mathbb{E}_{p_*(x,y)\bar{p}_*(\bar{x}_j^w|x,y)}[l_j(g_j(X^o), \overline{X}_j^w))]$ represents the expected risk of $g_j$ computed using $\overline{X}_j^w$. Additionally, $\widehat{\overline{R}}_{l_j}$ de-

notes the empirical risk, which approximates $\overline{R}_{l_j}$ by taking the sample average over $\overline{S}_j$. We assume that $\overline{R}_{l_j}$ satisfies either $R_{01,j}(g_j) = \overline{R}_{l_j}(g_j)$ or $R_{01,j}(g_j) \leq \overline{R}_{l_j}(g_j)$ for any $g_j$, or that the optimal solutions of $\overline{R}_{l_j}$ coincide with those of $R_{01,j}$. For any $\boldsymbol{r} = (r_1, \ldots, r_{F^{\mathrm{w}}}) \in \mathbb{R}_+^{F^{\mathrm{w}}}$, define the following hypothesis class:

$$\mathcal{G}(\boldsymbol{r}, \overline{S}) := \mathcal{G}_1(r_1, \overline{S}_1) \times \cdots \times \mathcal{G}_{F^{\mathrm{w}}}(r_{F^{\mathrm{w}}}, \overline{S}_{F^{\mathrm{w}}}),$$

where, $\mathcal{G}_j(r_j, \overline{S}_j) := \{g_j | g_j \in \mathcal{G}_j \wedge \widehat{\overline{R}}_{l_j}(g_j) \leq r_j\}, \forall j \in [F^{\mathrm{w}}]$. To explicitly indicate that $R_{l,\boldsymbol{g}}(f)$ is part of the objective function of $\boldsymbol{g}$, we denote it as $R_{l,f}(\boldsymbol{g})(\equiv R_{l,\boldsymbol{g}}(f))$ and the empirical risk of $R_{l,f}(\boldsymbol{g})$ is defined as $\widehat{R}_{l,f}(\boldsymbol{g})(\equiv \widehat{R}_{l,\boldsymbol{g}}(f))$. By performing the learning of $\boldsymbol{g}$ in LAC-dWFL's step (iii) as outlined below, the analysis of minimizing $R_{l,f}$ while reducing $\sum_{j \in [F^{\mathrm{w}}]} R_{01,j}$ becomes feasible:

$$\boldsymbol{g}_{f,\overline{S}}^{(\boldsymbol{r})} := \arg\min_{\boldsymbol{g} \in \mathcal{G}(\boldsymbol{r}, \overline{S})} \widehat{R}_{l,f}(\boldsymbol{g}). \qquad (4.9)$$

Based on the above definition, the assumptions and Lemma 4.1, the error bound for $\boldsymbol{g}_{f,\overline{S}}^{(\boldsymbol{r})}$ is presented in the following theorem. The proof is provided in Appendix A.5.

**Theorem 4.4.** *Suppose $S$ and $\overline{S}$ represent an ordinary dataset and a weak dataset of $n$ samples, respectively. Then, for any measurable $f \in \mathcal{F}$, $l$ bounded by $U_l < \infty$ and $\delta \in (0,1)$, the following holds with probability at least $1 - \delta$:*

$$R_{l,f}(\boldsymbol{g}_{f,\overline{S}}^{(\boldsymbol{r})}) - R_l(f) \leq$$
$$\left( 4\mathfrak{R}_n^*(\widetilde{\mathcal{G}}_{l,f}(\boldsymbol{r}, \overline{\boldsymbol{S}})) + 2U_l \sqrt{\frac{\log(2/\delta)}{2n}} \right)$$
$$+ \left\{ 2\sqrt{R_l(f)} + \left( 2U_l \sum_{j \in [F^{\mathrm{w}}]} R_{01,j}(g_{\overline{S},j}^{(r_j)}) \right)^{\frac{1}{2}} \right\} \qquad (4.10)$$
$$\times \left( 2U_l \sum_{j \in [F^{\mathrm{w}}]} R_{01,j}(g_{\overline{S},j}^{(r_j)}) \right)^{\frac{1}{2}}$$

*Here, $\widetilde{\mathcal{G}}_{l,f}(\boldsymbol{r}, \overline{S}) := \{(\boldsymbol{x}^{\mathrm{o}}, y) \mapsto l(f(\boldsymbol{g}(\boldsymbol{x}^{\mathrm{o}}), \boldsymbol{x}^{\mathrm{o}}), y) : \boldsymbol{g} \in \mathcal{G}(\boldsymbol{r}, \overline{S})\}$.*

The term $R_{01,j}(g_{\overline{S},j}^{(r_j)})$ in Eq. (4.10) can be further upper-bounded by defining $\mathcal{G}_j(r_j, \overline{S}_j)$ as the set of empirical risk minimizers obtained via weakly supervised learning using $\overline{S}_j$. For instance, when $\overline{X}_j^{\mathrm{w}}$ is a CF, and every hypothesis in $\mathcal{G}_j(r_j, \overline{S}_j)$ is an empirical risk minimizer obtained using the method of Ishida et al. (Ishida et al., 2017) and $\mathcal{G}_j$ is sufficiently large to satisfy $\min_{g_j \in \mathcal{G}_j} R_{01,j}(g_j) = 0$, then $R_{01,j}(g_{\overline{S},j}^{(r_j)})$ can be upper-bounded by Eq. (4.8). Therefore, when we define $\mathcal{G}_j(r_j, \overline{S}_j)$ as the class of empirical risk minimizers obtained by such a method with guaranteed consistency, and assume that the order of $\mathfrak{R}_n^*(\mathcal{G}_j)$ is $\mathcal{O}_p(1/n^{1/2})$, the order of the upper bound of $R_{01,j}(g_{\overline{S},j}^{(r_j)})$ is

also $\mathcal{O}_p(1/n^{1/2})$. Furthermore, assuming that $\mathfrak{R}_n^*(\mathcal{G}_j) \to 0$ as $n \to \infty$, it follows that $R_{01,j}(g_{\overline{S},j}^{(r_j)}) \to 0$ as $n \to \infty$.

In the following discussion, we assume that $R_{01,j}(g_{\overline{S},j}^{(r_j)})$ can be upper-bounded by a probability inequality of a form similar to that of Eq. (4.8).

Theorem 4.4 elucidates the influence of $f$'s prediction error and characteristics on the error bound for learning $\boldsymbol{g}$ via LAC-dWFL's step (iii), as well as the convergence of $\boldsymbol{g}$'s learning. First, Theorem 4.4 demonstrates that the convergence rate of the upper bound of $R_{l,f}(\boldsymbol{g}_{f,S}^{(\boldsymbol{r})})$ with respect to $n$ significantly depends on the expected risk $R_l(f)$. Specifically, assuming the orders of the Rademacher complexities in Eq. (4.10) are $\mathcal{O}_p(1/n^{1/2})$, the orders of the first and second terms on the RHS of Eq. (4.10) are $\mathcal{O}_p(1/n^{1/2})$ and $\mathcal{O}_p(1/n^{1/4})$, respectively. Consequently, the slower-decreasing second term becomes dominant when $R_l(f)$ is large. Although $R_l(f)$ cannot be directly minimized in WFL, it has been shown that minimizing $R_{l,\lambda}^{\mathrm{dWFL}}(f, \cdot)$ helps reduce $R_l(f)$ (Theorem 3.1). Therefore $R_l(f)$ is expected to decrease through LAC-dWFL's step (ii). Additionally, if $l$ is $L_l$-Lipschitz continuous and $f$ is $L_f$-Lipschitz continuous, then $\mathfrak{R}_n^*(\widetilde{\mathcal{G}}_{l,f}(\boldsymbol{r}, \boldsymbol{S})) \leq L_l L_f \mathfrak{R}_n^*(\mathcal{G}(\boldsymbol{r}, \boldsymbol{S}))$ holds (Mohri et al., 2018). This implies that the learning efficiency of $\boldsymbol{g}$ improves when $f$ exhibits smoother variation with respect to its input. Furthermore, combining Theorem 4.4 with Theorem 4.2 reveals the conditions under which iterative learning in LAC-dWFL achieves consistency. The proof is shown in Appendix A.6.

**Theorem 4.5.** *In addition to the conditions stated in Theorem 4.3, assume $f$ obtained via LAC-dWFL's step (ii) is Lipschitz continuous, and the Rademacher complexities about $\boldsymbol{g}$ asymptotically converge to $0$ as $n$ increases. Furthermore, for any $j \in [F^{\mathrm{w}}]$, define $\mathcal{G}_j(r_j, \overline{S}_j)$ as the set of empirical risk minimizers obtained by methods that use $\overline{S}_j$ and are guaranteed to achieve consistency. Then iterative learning achieves consistency.*

## 5. Experiments

In Section 4 we revealed two key properties of LAC-dWFL: (1) the mutual influence between the feature estimation models $\boldsymbol{g}$ and the label prediction model $f$ during their respective learning processes, and (2) the relationship between the generalization error and the number of training samples $n$ in WFL. Furthermore, the theoretical analysis demonstrates that sequential learning alone suffices for WFL. To validate the critical aspect of WFL, namely the impact of the estimation error of $\boldsymbol{g}$ on the learning of $f$ (Theorem 4.2), we evaluate how varying the estimation errors of $\boldsymbol{g}$ affects $f$'s learning performance.

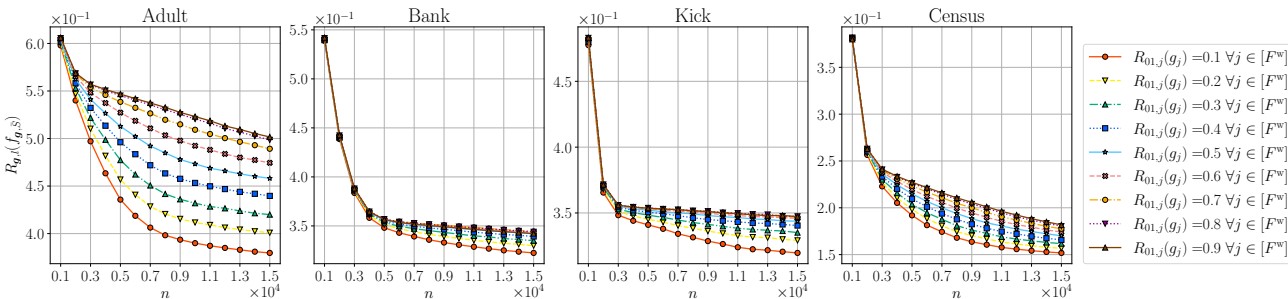

Figure 5.1. The relationship between the number of training samples $n$, $R_{l,g}(f_{g,\overline{S}})$, and various estimation errors of $g$.

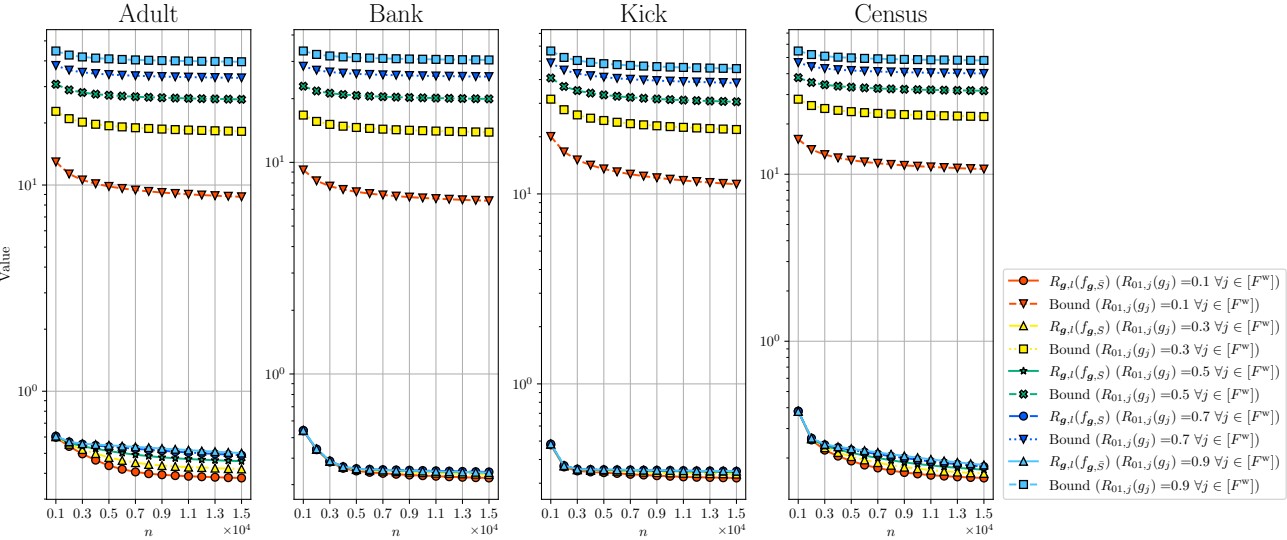

Figure 5.2. A comparison between $R_{l,g}(f_{g,\overline{S}})$ and the error bound derived in Theorem 4.2, for various estimation errors of $g$.

### 5.1. Experimental Settings

We used four real-world datasets: *Adult* (Becker & Kohavi, 1996), *Bank Marketing* (Moro & Cortez, 2014), *kick* (Vanschoren et al., 2013), and *Census-Income (KDD)* (cen, 2000; Dua & Graff, 2017). We refer to them as *Adult*, *Bank*, *Kick*, and *Census*, respectively. Details of these datasets are summarized in Appendix B.1. For each dataset, 50% of the samples were reserved as test data to estimate the generalization error. In this experiment, we focused on a representative case of WFs, where all categorical features are treated as CFs (Sugiyama & Uchida, 2024). Both the feature estimation models $g$ and the label prediction model $f$ were implemented using two-layer perceptrons with hidden layers of width 500 and ReLU as an activation function. Logistic loss was used as $l$. The Rademacher complexity, required for calculating the error bounds, was estimated using the method proposed by Neyshabur et al. (Neyshabur et al., 2015). Details of the experimen-

tal settings are summarized in Appendix B.2. The following results are the average of 5 trials. The experimental scripts used in this paper are available at the following URL: https://github.com/KOHsEMP/discrete_WFL

### 5.2. Impact of $g$ on $f$'s learning

This section investigates how the estimation errors of feature estimation models $g$ affects the learning of the label prediction model $f$. To perform this investigation, precise control over the estimation errors of $g$ is required. However, achieving such fine-grained control through WSL methods is challenging. Importantly, for this experiment, the method used to obtain $g$ is less relevant than the influence of its estimation errors on $f$. Thus, synthetic estimation functions for $g$ are employed, which randomly misestimate with controlled error rates, enabling systematic examination of the impact of $g$'s estimation errors on $f$'s learning.

We vary the estimation error of $\boldsymbol{g}$ from 10% to 90% and train $f$ under these settings. Figure 5.1 shows the relationship between $n$ and $R_{l,\boldsymbol{g}}(f_{\boldsymbol{g},\overline{S}})$. The results confirm that, as shown in Theorem 4.2, lower estimation errors of $\boldsymbol{g}$ lead to a higher reduction rate of $R_{l,\boldsymbol{g}}(f_{\boldsymbol{g},\overline{S}})$ as $n$ increases.

Additionally, Figure 5.2 compares the generalization error of $f$, shown in Figure 5.1, with the error bound in Theorem 4.2. Figure 5.2 shows that the decrease in $R_{l,\boldsymbol{g}}(f_{\boldsymbol{g},\overline{S}})$ and its bound with increasing $n$ exhibits a similar trend, with the reduction becoming more significant as the estimation error of $\boldsymbol{g}$. Therefore, Theorem 4.2 effectively captures a fundamental characteristic of WFL, specifically the influence of $\boldsymbol{g}$ on the actual learning of $f$, which is consistent across various scenarios. The discrepancy between $R_{l,\boldsymbol{g}}(f_{\boldsymbol{g},\overline{S}})$ and our bound in Figure 5.2 can be attributed to the fact that our bound does not account for the feature importance of WFs in predicting $Y$. This suggests a new research direction for deriving error bounds that incorporate the feature importance of WFs. Our results are considered to provide a critical foundation for such an approach.

## 6. Conclusion

This presented a unified formalization and theoretical analysis of discrete WFL. First, we proposed a formulation of WFL capable of handling arbitrary discrete WFs. We validated this formulation by demonstrating that the introduced objective function aids in learning a label prediction model $f$ that captures the true input-output relationship. Within this framework, we performed a generalization error analysis for LAC-dWFL, a generalized learning algorithm class designed to learn both feature estimation models $\boldsymbol{g}$ and $f$. This analysis revealed the detailed influence of the estimation errors of $\boldsymbol{g}$ and $f$ on the error bounds of $f$ and $\boldsymbol{g}$, respectively. Additionally, we identified theoretical conditions under which consistency can be achieved for the sequential and iterative learning approaches in LAC-dWFL. Finally, numerical experiments on real-world datasets verified that our theoretical results align with observed learning behavior. This study provides comprehensive theoretical insights into various problem settings, such as ItR and CF, involving discrete WFs.

## Impact Statement

Our paper presents a formalization and theoretical analysis of discrete WFL that accommodates arbitrary discrete WFs. Understanding the impact of low-quality input features on the training of predictive models is crucial for the development and deployment of safe machine learning systems. This is because degraded input feature quality can potentially lead to predictive models that are vulnerable to adversarial attacks or that propagate socially undesirable biases. The theoretical results presented in this work are expected to provide a foundational perspective for discussions concerning the interplay between input feature quality and the safety of machine learning.

## Acknowledgment

This work was supported in part by the Japan Society for the Promotion of Science through Grants-in-Aid for Scientific Research (C) (23K11111).

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

# A. Proofs

## A.1. Proof of Theorem 3.1

*Proof of Theorem 3.1.* For any $f \in \mathcal{F}$, $g \in \mathcal{G}$, and $l$ bounded by $U_l < \infty$, the following inequality holds:

$$
\begin{aligned}
R_l&(f) - R_{l,g}(f) \\
&= \mathbb{E}_{p_*(\boldsymbol{x},y)}[l(f(\boldsymbol{X}),Y)] - \mathbb{E}_{p_*(\boldsymbol{x}^\circ,y)q_g(\boldsymbol{x}^\mathrm{w}|\boldsymbol{x}^\circ)}[l(f(\boldsymbol{X}),Y)] \\
&= \mathbb{E}_{p_*(\boldsymbol{x}^\circ,y)}\left[ \sum_{\boldsymbol{x}^\mathrm{w}\in\mathcal{X}^\mathrm{w}} l(f(\boldsymbol{X}^\circ,\boldsymbol{x}^\mathrm{w}),Y)(p_*(\boldsymbol{x}^\mathrm{w}|\boldsymbol{X}^\circ,Y) - q_g(\boldsymbol{x}^\mathrm{w}|\boldsymbol{X}^\circ)) \right] \\
&= \mathbb{E}_{p_*(\boldsymbol{x}^\circ,y)}\left[ \sum_{\boldsymbol{x}^\mathrm{w}\in\mathcal{X}^\mathrm{w}} l(f(\boldsymbol{X}^\circ,\boldsymbol{x}^\mathrm{w}),Y)(p_*(\boldsymbol{x}^\mathrm{w}|\boldsymbol{X}^\circ,Y) - \mathbb{1}_{[\boldsymbol{x}^\mathrm{w}=g(\boldsymbol{X}^\circ)]}) \right] \\
&\leq \mathbb{E}_{p_*(\boldsymbol{x}^\circ,y)}\left[ \sum_{\boldsymbol{x}^\mathrm{w}\in\mathcal{X}^\mathrm{w}} l(f(\boldsymbol{X}^\circ,\boldsymbol{x}^\mathrm{w}),Y)(p_*(\boldsymbol{x}^\mathrm{w}|\boldsymbol{X}^\circ,Y) - \mathbb{1}_{[\boldsymbol{x}^\mathrm{w}=g(\boldsymbol{X}^\circ)]}p_*(\boldsymbol{x}^\mathrm{w}|\boldsymbol{X}^\circ,Y)) \right] \\
&= \mathbb{E}_{p_*(\boldsymbol{x}^\circ,y)}\left[ \sum_{\boldsymbol{x}^\mathrm{w}\in\mathcal{X}^\mathrm{w}} l(f(\boldsymbol{X}^\circ,\boldsymbol{x}^\mathrm{w}),Y)p_*(\boldsymbol{x}^\mathrm{w}|\boldsymbol{X}^\circ,Y)(1 - \mathbb{1}_{[\boldsymbol{x}^\mathrm{w}=g(\boldsymbol{X}^\circ)]}) \right] \\
&\leq \mathbb{E}_{p_*(\boldsymbol{x},y)}\left[ l(f(\boldsymbol{X}^\circ,\boldsymbol{X}^\mathrm{w}),Y) \sum_{j\in[F^\mathrm{w}]} (1 - \mathbb{1}_{[X_j^\mathrm{w}=g_j(\boldsymbol{X}^\circ)]}) \right] \\
&= \mathbb{E}_{p_*(\boldsymbol{x},y)}\left[ l(f(\boldsymbol{X}^\circ,\boldsymbol{X}^\mathrm{w}),Y) \sum_{j\in[F^\mathrm{w}]} l_{01}(g_j(\boldsymbol{X}^\circ),X_j^\mathrm{w}) \right] \\
&\leq U_l \mathbb{E}_{p_*(\boldsymbol{x},y)}\left[ \sum_{j\in[F^\mathrm{w}]} l_{01}(g_j(\boldsymbol{X}^\circ),X_j^\mathrm{w}) \right].
\end{aligned}
$$

The second inequality arises from the decomposition of the 0-1 loss. This decomposition is intended to derive the risk for each $g_j$. The final inequality uses the assumption that the maximum value of the loss function $l$ is $U_l$. Therefore, Theorem 3.1 is proved. $\qquad\square$

## A.2. Proof of Lemma 4.1

*Proof of Lemma 4.1.* The LHS of Eq. (4.6) can be rewritten as follows:

$$
\begin{aligned}
|R_l&(f) - R_{l,g}(f)| \\
&= |\mathbb{E}_{p_*(\boldsymbol{x},y)}[l(f(\boldsymbol{X}),Y)] - \mathbb{E}_{p_*(\boldsymbol{x}^\circ,y)q_g(\boldsymbol{x}^\mathrm{w}|\boldsymbol{x}^\circ)}[l(f(\boldsymbol{X}),Y)]| \\
&= \left| \mathbb{E}_{p_*(\boldsymbol{x}^\circ,y)}\left[ \sum_{\boldsymbol{x}^\mathrm{w}\in\mathcal{X}^\mathrm{w}} l(f(\boldsymbol{x}^\mathrm{w},\boldsymbol{X}^\circ),Y)\{p_*(\boldsymbol{x}^\mathrm{w}|\boldsymbol{X}^\circ,Y) - q_g(\boldsymbol{x}^\mathrm{w}|\boldsymbol{X}^\circ)\} \right] \right| \\
&= \left| \mathbb{E}_{p_*(\boldsymbol{x}^\circ,y)}\left[ \sum_{\boldsymbol{x}^\mathrm{w}\in\mathcal{X}^\mathrm{w}} l(f(\boldsymbol{x}^\mathrm{w},\boldsymbol{X}^\circ),Y)\{p_*(\boldsymbol{x}^\mathrm{w}|\boldsymbol{X}^\circ,Y) - \mathbb{1}_{[\boldsymbol{x}^\mathrm{w}=g(\boldsymbol{X}^\circ)]}\} \right] \right| \\
&= \left| \mathbb{E}_{p_*(\boldsymbol{x}^\circ,y)}\left[ \sum_{\boldsymbol{x}^\mathrm{w}\in\mathcal{X}^\mathrm{w}} \left\{ l(f(\boldsymbol{x}^\mathrm{w},\boldsymbol{X}^\circ),Y)p_*(\boldsymbol{x}^\mathrm{w}|\boldsymbol{X}^\circ,Y) - l(f(\boldsymbol{x}^\mathrm{w},\boldsymbol{X}^\circ),Y)\mathbb{1}_{[\boldsymbol{x}^\mathrm{w}=g(\boldsymbol{X}^\circ)]} \right\} \right] \right| \\
&= \left| \mathbb{E}_{p_*(\boldsymbol{x}^\circ,y)}\left[ \sum_{\boldsymbol{x}^\mathrm{w}\in\mathcal{X}^\mathrm{w}} \left\{ l(f(\boldsymbol{x}^\mathrm{w},\boldsymbol{X}^\circ),Y)p_*(\boldsymbol{x}^\mathrm{w}|\boldsymbol{X}^\circ,Y) - l(f(\boldsymbol{x}^\mathrm{w},\boldsymbol{X}^\circ),Y)\mathbb{1}_{[\boldsymbol{x}^\mathrm{w}=g(\boldsymbol{X}^\circ)]}p_*(\boldsymbol{x}^\mathrm{w}|\boldsymbol{X}^\circ,Y) \right. \right. \right. \right. \\
&\qquad\qquad \left. \left. \left. \left. + l(f(\boldsymbol{x}^\mathrm{w},\boldsymbol{X}^\circ),Y)\mathbb{1}_{[\boldsymbol{x}^\mathrm{w}=g(\boldsymbol{X}^\circ)]}p_*(\boldsymbol{x}^\mathrm{w}|\boldsymbol{X}^\circ,Y) - l(f(\boldsymbol{x}^\mathrm{w},\boldsymbol{X}^\circ),Y)\mathbb{1}_{[\boldsymbol{x}^\mathrm{w}=g(\boldsymbol{X}^\circ)]} \right\} \right] \right| \right.
\end{aligned}
$$

$$\leq \mathbb{E}_{p_*(\boldsymbol{x}^\mathrm{o}, y)} \left[ \underbrace{\sum_{\boldsymbol{x}^\mathrm{w} \in \mathcal{X}^\mathrm{w}} \left| l(f(\boldsymbol{x}^\mathrm{w}, \boldsymbol{X}^\mathrm{o}), Y) p_*(\boldsymbol{x}^\mathrm{w} | \boldsymbol{X}^\mathrm{o}, Y) - l(f(\boldsymbol{x}^\mathrm{w}, \boldsymbol{X}^\mathrm{o}), Y) \mathbb{1}_{[\boldsymbol{x}^\mathrm{w} = \boldsymbol{g}(\boldsymbol{X}^\mathrm{o})]} p_*(\boldsymbol{x}^\mathrm{w} | \boldsymbol{X}^\mathrm{o}, Y) \right|}_{\text{(a1)}} \right.$$

$$\left. + \underbrace{\sum_{\boldsymbol{x}^\mathrm{w} \in \mathcal{X}^\mathrm{w}} \left| l(f(\boldsymbol{x}^\mathrm{w}, \boldsymbol{X}^\mathrm{o}), Y) \mathbb{1}_{[\boldsymbol{x}^\mathrm{w} = \boldsymbol{g}(\boldsymbol{X}^\mathrm{o})]} p_*(\boldsymbol{x}^\mathrm{w} | \boldsymbol{X}^\mathrm{o}, Y) - l(f(\boldsymbol{x}^\mathrm{w}, \boldsymbol{X}^\mathrm{o}), Y) \mathbb{1}_{[\boldsymbol{x}^\mathrm{w} = \boldsymbol{g}(\boldsymbol{X}^\mathrm{o})]} \right|}_{\text{(a2)}} \right]. \qquad \text{(A.11)}$$

The term (a1) in Eq. (A.11) can be expressed as:

$$\text{(a1)} = \sum_{\boldsymbol{x}^\mathrm{w} \in \mathcal{X}^\mathrm{w}} l(f(\boldsymbol{x}^\mathrm{w}, \boldsymbol{X}^\mathrm{o}), Y) p_*(\boldsymbol{x}^\mathrm{w} | \boldsymbol{X}^\mathrm{o}, Y)(1 - \mathbb{1}_{[\boldsymbol{x}^\mathrm{w} = \boldsymbol{g}(\boldsymbol{X}^\mathrm{o})]})$$

$$= \mathbb{E}_{p_*(\boldsymbol{x}^\mathrm{w} | \boldsymbol{X}^\mathrm{o}, Y)} \left[ l(f(\boldsymbol{X}^\mathrm{w}, \boldsymbol{X}^\mathrm{o}), Y) l_{01}(\boldsymbol{g}(\boldsymbol{X}^\mathrm{o}), \boldsymbol{X}^\mathrm{w}) \right].$$

Here, $l_{01}(\boldsymbol{g}(\boldsymbol{X}^\mathrm{o}), \boldsymbol{X}^\mathrm{w}) := 1 - \mathbb{1}_{[\boldsymbol{X}^\mathrm{w} = \boldsymbol{g}(\boldsymbol{X}^\mathrm{o})]}$ The term (a2) in Eq. (A.11) can be expressed as:

$$\text{(a2)} = \sum_{\boldsymbol{x}^\mathrm{w} \in \mathcal{X}^\mathrm{w}} \left\{ l(f(\boldsymbol{x}^\mathrm{w}, \boldsymbol{X}^\mathrm{o}), Y) \mathbb{1}_{[\boldsymbol{x}^\mathrm{w} = \boldsymbol{g}(\boldsymbol{X}^\mathrm{o})]} - l(f(\boldsymbol{x}^\mathrm{w}, \boldsymbol{X}^\mathrm{o}), Y) \mathbb{1}_{[\boldsymbol{x}^\mathrm{w} = \boldsymbol{g}(\boldsymbol{X}^\mathrm{o})]} p_*(\boldsymbol{x}^\mathrm{w} | \boldsymbol{X}^\mathrm{o}, Y) \right\}$$

$$= l(f(\boldsymbol{g}(\boldsymbol{X}^\mathrm{o}), \boldsymbol{X}^\mathrm{o}), Y) - l(f(\boldsymbol{g}(\boldsymbol{X}^\mathrm{o}), \boldsymbol{X}^\mathrm{o}), Y) p_*(\boldsymbol{g}(\boldsymbol{X}^\mathrm{o}) | \boldsymbol{X}^\mathrm{o}, Y)$$

$$= l(f(\boldsymbol{g}(\boldsymbol{X}^\mathrm{o}), \boldsymbol{X}^\mathrm{o}), Y) \left( \sum_{\boldsymbol{x}^\mathrm{w} \in \mathcal{X}^\mathrm{w}} p_*(\boldsymbol{x}^\mathrm{w} | \boldsymbol{X}^\mathrm{o}, Y) \right)$$

$$- l(f(\boldsymbol{g}(\boldsymbol{X}^\mathrm{o}), \boldsymbol{X}^\mathrm{o}), Y) \left( \sum_{\boldsymbol{x}^\mathrm{w} \in \mathcal{X}^\mathrm{w}} p_*(\boldsymbol{x}^\mathrm{w} | \boldsymbol{X}^\mathrm{o}, Y) \mathbb{1}_{[\boldsymbol{x}^\mathrm{w} = \boldsymbol{g}(\boldsymbol{X}^\mathrm{o})]} \right)$$

$$= l(f(\boldsymbol{g}(\boldsymbol{X}^\mathrm{o}), \boldsymbol{X}^\mathrm{o}), Y) \left( \sum_{\boldsymbol{x}^\mathrm{w} \in \mathcal{X}^\mathrm{w}} p_*(\boldsymbol{x}^\mathrm{w} | \boldsymbol{X}^\mathrm{o}, Y)(1 - \mathbb{1}_{[\boldsymbol{X}^\mathrm{w} = \boldsymbol{g}(\boldsymbol{X}^\mathrm{o})]}) \right)$$

$$= l(f(\boldsymbol{g}(\boldsymbol{X}^\mathrm{o}), \boldsymbol{X}^\mathrm{o}), Y) \mathbb{E}_{p_*(\boldsymbol{x}^\mathrm{w} | \boldsymbol{X}^\mathrm{o}, Y)} \left[ l_{01}(\boldsymbol{g}(\boldsymbol{X}^\mathrm{o}), \boldsymbol{X}^\mathrm{w}) \right].$$

By substituting these results into (a1) and (a2) of Eq. (A.11), Eq. (A.11) can be rewritten as follows:

$$|R_l(f) - R_{l,\boldsymbol{g}}(f)| \leq \mathbb{E}_{p_*(\boldsymbol{x}, y)} \left[ \left( l(f(\boldsymbol{X}^\mathrm{w}, \boldsymbol{X}^\mathrm{o}), Y) + l(f(\boldsymbol{g}(\boldsymbol{X}^\mathrm{o}), \boldsymbol{X}^\mathrm{o}), Y) \right) l_{01}(\boldsymbol{g}(\boldsymbol{X}^\mathrm{o}), \boldsymbol{X}^\mathrm{w}) \right]. \qquad \text{(A.12)}$$

Since $l$, $l_{01}$, $f$ and $\boldsymbol{g}$ are all measurable functions, applying the Cauchy-Schwarz inequality to the RHS of Eq. (A.12), $|R_l(f) - R_{l,\boldsymbol{g}}(f)|$ can be upper-bounded as follows:

$$|R_l(f) - R_{l,\boldsymbol{g}}(f)|$$

$$\leq \mathbb{E}_{p_*(\boldsymbol{x}, y)} \left[ \left( l(f(\boldsymbol{X}^\mathrm{w}, \boldsymbol{X}^\mathrm{o}), Y) + l(f(\boldsymbol{g}(\boldsymbol{X}^\mathrm{o}), \boldsymbol{X}^\mathrm{o}), Y) \right) l_{01}(\boldsymbol{g}(\boldsymbol{X}^\mathrm{o}), \boldsymbol{X}^\mathrm{w}) \right]$$

$$\leq \left( \underbrace{\mathbb{E}_{p_*(\boldsymbol{x}, y)} \left[ \left( l(f(\boldsymbol{X}^\mathrm{w}, \boldsymbol{X}^\mathrm{o}), Y) \right)^2 \right]}_{\text{(b1)}} \times \underbrace{\mathbb{E}_{p_*(\boldsymbol{x})} \left[ \left( l_{01}(\boldsymbol{g}(\boldsymbol{X}^\mathrm{o}), \boldsymbol{X}^\mathrm{w}) \right)^2 \right]}_{\text{(b2)}} \right)^{\frac{1}{2}} \qquad \text{(A.13)}$$

$$+ \left( \underbrace{\mathbb{E}_{p_*(\boldsymbol{x}^\mathrm{o}, y)} \left[ \left( l(f(\boldsymbol{g}(\boldsymbol{X}^\mathrm{o}), \boldsymbol{X}^\mathrm{o}), Y) \right)^2 \right]}_{\text{(b3)}} \times \underbrace{\mathbb{E}_{p_*(\boldsymbol{x})} \left[ \left( l_{01}(\boldsymbol{g}(\boldsymbol{X}^\mathrm{o}), \boldsymbol{X}^\mathrm{w}) \right)^2 \right]}_{\text{(b2)}} \right)^{\frac{1}{2}}.$$

The terms (b1)–(b3) of Eq. (A.13) can be expressed as:

$$(\text{b1}) = \mathbb{E}_{p_*(\boldsymbol{x},y)}\left[\left(l(f(\boldsymbol{X}),Y)\right)^2 - 0^2\right] \leq 2U_l \mathbb{E}_{p_*(\boldsymbol{x},y)}[l(f(\boldsymbol{X}),Y)] = 2U_l R_l(f),$$

$$(\text{b2}) = \mathbb{E}_{p_*(\boldsymbol{x})}[l_{01}(\boldsymbol{g}(\boldsymbol{X}^{\circ}),\boldsymbol{X}^{\mathrm{w}})] \leq \sum_{j\in[F^{\mathrm{w}}]} \mathbb{E}_{p_*(\boldsymbol{x})}[l_{01}(g_j(\boldsymbol{X}^{\circ}),X_j^{\mathrm{w}})] = \sum_{j\in[F^{\mathrm{w}}]} R_{01,j}(g_j),$$

$$(\text{b3}) = \mathbb{E}_{p_*(\boldsymbol{x}^{\circ},y)}\left[\left(l(f(\boldsymbol{g}(\boldsymbol{X}^{\circ}),\boldsymbol{X}^{\circ}),Y)\right)^2 - 0^2\right] \leq 2U_l \mathbb{E}_{p_*(\boldsymbol{x}^{\circ},y)}[l(f(\boldsymbol{g}(\boldsymbol{X}^{\circ}),\boldsymbol{X}^{\circ}),Y)] = 2U_l R_{l,\boldsymbol{g}}(f).$$

Here, in (b1) and (b3), the fact that the function $x \mapsto x^2$ is $2U_l$-Lipschitz continuous on the interval $[0,U_l]$ was utilized.

Applying the above inequalities related to (b1)–(b3) to the RHS of Eq. (A.13), $|R_l(f) - R_{l,\boldsymbol{g}}(f)|$ can be upper-bounded as follows:

$$
\begin{aligned}
|R_l&(f) - R_{l,\boldsymbol{g}}(f)| \\
&\leq \left\{ 2U_l R_l(f) \sum_{j\in[F^{\mathrm{w}}]} R_{01,j}(g_j) \right\}^{\frac{1}{2}} + \left\{ 2U_l R_{l,\boldsymbol{g}}(f) \sum_{j\in[F^{\mathrm{w}}]} R_{01,j}(g_j) \right\}^{\frac{1}{2}} \\
&= \left( \sqrt{R_l(f)} + \sqrt{R_{l,\boldsymbol{g}}(f)} \right) \left( 2U_l \sum_{j\in[F^{\mathrm{w}}]} R_{01,j}(g_j) \right)^{\frac{1}{2}}.
\end{aligned}
\tag{A.14}
$$

Thus, Lemma 4.1 is proven.

$\square$

### A.3. Proof of Theorem 4.2

From Lemma 4.1, the following lemma holds:

**Lemma A.1.** *For any $f \in \mathcal{F}$, $\boldsymbol{g} \in \mathcal{G}$ and $l$ bounded by $U_l < \infty$, the following inequality holds:*

$$|R_l(f) - R_{l,\boldsymbol{g}}(f)| \leq \left( 2\sqrt{R_l(f)} + \left( 2U_l \sum_{j\in[F^{\mathrm{w}}]} R_{01,j}(g_j) \right)^{\frac{1}{2}} \right) \left( 2U_l \sum_{j\in[F^{\mathrm{w}}]} R_{01,j}(g_j) \right)^{\frac{1}{2}}. \tag{A.15}$$

*Proof of Lemma A.1.* From Lemma 4.1, for any $f \in \mathcal{F}$, $\boldsymbol{g} \in \mathcal{G}$ and $l$ bounded by $U_l < \infty$, the following inequality holds:

$$\left| \sqrt{R_l(f)} - \sqrt{R_{l,\boldsymbol{g}}(f)} \right| \leq \left( 2U_l \sum_{j\in[F^{\mathrm{w}}]} R_{01,j}(g_j) \right)^{\frac{1}{2}}. \tag{A.16}$$

Hence, $\sqrt{R_{l,\boldsymbol{g}}(f)}$ can be upper-bounded as follows:

$$
\begin{aligned}
\sqrt{R_{l,\boldsymbol{g}}(f)} &= \sqrt{R_{l,\boldsymbol{g}}(f)} + \sqrt{R_l(f)} - \sqrt{R_l(f)} \\
&\leq \sqrt{R_l(f)} + \left| \sqrt{R_l(f)} - \sqrt{R_{l,\boldsymbol{g}}(f)} \right| \\
&\leq \sqrt{R_l(f)} + \left( 2U_l \sum_{j\in[F^{\mathrm{w}}]} R_{01,j}(g_j) \right)^{\frac{1}{2}}.
\end{aligned}
\tag{A.17}
$$

By applying the above inequality to the RHS of Eq. (A.14), $|R_l(f) - R_{l,\boldsymbol{g}}(f)|$ can be upper-bounded as follows:

$$|R_l(f) - R_{l,\boldsymbol{g}}(f)| \le \left(2\sqrt{R_l(f)} + \left(2U_l \sum_{j \in [F^{\mathrm{w}}]} R_{01,j}(g_j)\right)^{\frac{1}{2}}\right)\left(2U_l \sum_{j \in [F^{\mathrm{w}}]} R_{01,j}(g_j)\right)^{\frac{1}{2}}. \tag{A.18}$$

$\square$

By leveraging Lemma A.1, Theorem 4.2 is proven as follows.

*Proof of Theorem 4.2.* From Section 3.1, we define the empirical risk minimizer in ordinary supervised learning as follows:

$$f_S := \arg\min_{f \in \mathcal{F}} \widehat{R}_l(f).$$

The LHS of Eq. (4.7) can be rewritten as:

$$
\begin{aligned}
&R_{l,\boldsymbol{g}}(f_{\boldsymbol{g},\overline{S}}) - R_l(f_\mathcal{F}) \\
&= \underbrace{R_{l,\boldsymbol{g}}(f_{\boldsymbol{g},\overline{S}}) - \widehat{R}_{l,\boldsymbol{g}}(f_{\boldsymbol{g},\overline{S}})}_{\text{(a1)}} + \underbrace{\widehat{R}_{l,\boldsymbol{g}}(f_{\boldsymbol{g},\overline{S}}) - R_{l,\boldsymbol{g}}(f_S)}_{\text{(a2)}} \\
&\quad + \underbrace{R_{l,\boldsymbol{g}}(f_S) - R_l(f_S)}_{\text{(a3)}} + \underbrace{R_l(f_S) - R_l(f_\mathcal{F})}_{\text{(a4)}}.
\end{aligned}
\tag{A.19}
$$

The terms (a1) and (a2) in Eq. (A.19) can be upper-bounded as follows:

$$(\text{a1}) \le \max_{f \in \mathcal{F}} |R_{l,\boldsymbol{g}}(f) - \widehat{R}_{l,\boldsymbol{g}}(f)|,$$

$$(\text{a2}) \le \widehat{R}_{l,\boldsymbol{g}}(f_S) - R_{l,\boldsymbol{g}}(f_S) \le \max_{f \in \mathcal{F}} |R_{l,\boldsymbol{g}}(f) - \widehat{R}_{l,\boldsymbol{g}}(f)|.$$

The term (a3) in Eq. (A.19) can be upper-bounded using Lemma A.1 as follows:

$$(\text{a3}) \le \left\{2\sqrt{R_l(f_S)} + \left(2U_l \sum_{j \in [F^{\mathrm{w}}]} R_{01,j}(g_j)\right)^{\frac{1}{2}}\right\}\left(2U_l \sum_{j \in [F^{\mathrm{w}}]} R_{01,j}(g_j)\right)^{\frac{1}{2}}. \tag{A.20}$$

Additionally, $R_l(f_S)$ can be upper-bounded as follows:

$$
\begin{aligned}
R_l(f_S) &= R_l(f_S) - \widehat{R}_l(f_S) + \widehat{R}_l(f_S) - R_l(f_\mathcal{F}) + R_l(f_\mathcal{F}) \\
&\le R_l(f_S) - \widehat{R}_l(f_S) + \widehat{R}_l(f_\mathcal{F}) - R_l(f_\mathcal{F}) + R_l(f_\mathcal{F}) \\
&\le R_l(f_\mathcal{F}) + 2\max_{f \in \mathcal{F}} |R_l(f) - \widehat{R}_l(f)|.
\end{aligned}
\tag{A.21}
$$

Hence, (a3) in Eq. (A.19) can be upper-bounded as follows:

$$(\text{a3}) \le \left(2\left(R_l(f_\mathcal{F}) + 2\max_{f \in \mathcal{F}} |R_l(f) - \widehat{R}_l(f)|\right)^{\frac{1}{2}} + \left(2U_l \sum_{j \in [F^{\mathrm{w}}]} R_{01,j}(g_j)\right)^{\frac{1}{2}}\right)\left(2U_l \sum_{j \in [F^{\mathrm{w}}]} R_{01,j}(g_j)\right)^{\frac{1}{2}}. \tag{A.22}$$

Similarly, the term (a4) in Eq. (A.19) can be upper-bounded as follows:

$$R_l(f_S) - R_l(f_\mathcal{F}) \le 2\max_{f \in \mathcal{F}} |R_l(f) - \widehat{R}_l(f)|.$$

By applying the above inequalities regarding (a1)–(a4) to the RHS of Eq. (A.19), it can be upper-bounded as follows:

$$
\begin{aligned}
&R_{l,\boldsymbol{g}}(f_{\boldsymbol{g},\overline{S}}) - R_l(f_\mathcal{F}) \\
&\le 2\max_{f \in \mathcal{F}} |R_{l,\boldsymbol{g}}(f) - \widehat{R}_{l,\boldsymbol{g}}(f)| + 2\max_{f \in \mathcal{F}} |R_l(f) - \widehat{R}_l(f)| \\
&\quad + \left\{2\left(R_l(f_\mathcal{F}) + 2\max_{f \in \mathcal{F}} |R_l(f) - \widehat{R}_l(f)|\right)^{\frac{1}{2}} + \left(2U_l \sum_{j \in [F^{\mathrm{w}}]} R_{01,j}(g_j)\right)^{\frac{1}{2}}\right\}\left(2U_l \sum_{j \in [F^{\mathrm{w}}]} R_{01,j}(g_j)\right)^{\frac{1}{2}}.
\end{aligned}
\tag{A.23}
$$

From the uniform law of large numbers (Mohri et al., 2018), for any $\delta \in (0,1)$, the following holds with a probability of at least $1 - \delta/2$:

$$\max_{f \in \mathcal{F}} |R_{l,\boldsymbol{g}}(f) - \widehat{R}_{l,\boldsymbol{g}}(f)| \leq 2\mathfrak{R}_n^{\boldsymbol{g}}(\widetilde{\mathcal{F}}_l) + U_l \sqrt{\frac{\log(4/\delta)}{2n}},$$

$$\max_{f \in \mathcal{F}} |R_l(f) - \widehat{R}_l(f)| \leq 2\mathfrak{R}_n^*(\widetilde{\mathcal{F}}_l) + U_l \sqrt{\frac{\log(4/\delta)}{2n}}.$$

Here, $\widetilde{\mathcal{F}}_l := \{(\boldsymbol{x}, y) \mapsto l(f(\boldsymbol{x}), y) : f \in \mathcal{F}\}$.

From the assumption that $l$ is $L_l$-Lipschitz continuous, it follows that $\mathfrak{R}_n^*(\widetilde{\mathcal{F}}_l) \leq L_l \mathfrak{R}_n^*(\mathcal{F})$ and $\mathfrak{R}_n^{\boldsymbol{g}}(\widetilde{\mathcal{F}}_l) \leq L_l \mathfrak{R}_n^{\boldsymbol{g}}(\mathcal{F})$ (Lemma 26.9 in (Shalev-Shwartz & Ben-David, 2014)).

Thus, for any $\delta \in (0,1)$, the following holds with a probability of at least $1 - \delta$:

$$R_{l,\boldsymbol{g}}(f_{\boldsymbol{g},\overline{S}}) - R_l(f_{\mathcal{F}})$$

$$\leq 4 \left( L_l \mathfrak{R}_n^*(\mathcal{F}) + L_l \mathfrak{R}_n^{\boldsymbol{g}}(\mathcal{F}) + U_l \sqrt{\frac{\log(4/\delta)}{2n}} \right)$$

$$+ \left\{ 2 \left( R_l(f_{\mathcal{F}}) + 4L_l \mathfrak{R}_n^*(\mathcal{F}) + 2U_l \sqrt{\frac{\log(4/\delta)}{2n}} \right)^{\frac{1}{2}} + \left( 2U_l \sum_{j \in [F^{\mathrm{w}}]} R_{01,j}(g_j) \right)^{\frac{1}{2}} \right\} \left( 2U_l \sum_{j \in [F^{\mathrm{w}}]} R_{01,j}(g_j) \right)^{\frac{1}{2}}.$$

(A.24)

$\square$

## A.4. Proof of Theorem 4.3

*Proof of Theorem 4.3.* By assumption, there exist true deterministic functions $g_j^* : \mathcal{X}^{\mathrm{o}} \to \mathcal{X}_j^{\mathrm{w}}$ for any $j \in [F^{\mathrm{w}}]$, and $(g_1^*, \ldots, g_{F^{\mathrm{w}}}^*) \in \mathcal{G}$. Therefore, when $\boldsymbol{g}_{\overline{S}} = (g_{\overline{S},1}, \ldots, g_{\overline{S},F^{\mathrm{w}}})$ is obtained by the methods that achieve consistency (Cour et al., 2011; Feng et al., 2020; Xu et al., 2021; Natarajan et al., 2013; Ishida et al., 2017; Yu et al., 2018), the following holds:

$$n \to \infty, \quad R_{01,j}(g_{\overline{S},j}) \to 0, \quad \forall j \in [F^{\mathrm{w}}].$$

(A.25)

Additionally, by assumption, there exists a true deterministic function $f^* : \mathcal{X} \to \mathcal{Y}$ for label prediction, and $f^* \in \mathcal{F}$. Hence, the following holds:

$$R_l(f_{\mathcal{F}}) = 0.$$

(A.26)

Thus, if $\mathfrak{R}_n^*(\mathcal{F})$ and $\mathfrak{R}_n^{\boldsymbol{g}}(\mathcal{F})$ are monotonically decreasing with respect to $n$ and converge to 0, the error bound established in Theorem 4.2 converges to 0 as $n \to \infty$.

$\square$

## A.5. Proof of Theorem 4.4

For a weak dataset $\overline{S}$ and a positive real-valued vector $\boldsymbol{r}$, define the feature estimation models $\boldsymbol{g}_{\overline{S}}^{(\boldsymbol{r})} = (g_{\overline{S},1}^{(r_1)}, \ldots, g_{\overline{S},F^{\mathrm{w}}}^{(r_{F^{\mathrm{w}}})})$ as follows:

$$g_{\overline{S},j}^{(r_j)} := \arg \min_{g \in \mathcal{G}(r_j, \overline{S})} \widehat{\overline{R}}_{l_j}(g), \quad \forall j \in [F^{\mathrm{w}}].$$

(A.27)

Using Lemma A.1, Theorem 4.4 is proven as follows.

*Proof of Theorem4.4.* The LHS of Eq. (4.10) can be rewritten as follows:

$$R_{l,f}(\boldsymbol{g}_{f,\overline{S}}^{(\boldsymbol{r})}) - R_l(f) = \underbrace{R_{l,f}(\boldsymbol{g}_{f,\overline{S}}^{(\boldsymbol{r})}) - \widehat{R}_{l,f}(\boldsymbol{g}_{f,\overline{S}}^{(\boldsymbol{r})})}_{(a1)} + \underbrace{\widehat{R}_{l,f}(\boldsymbol{g}_{f,\overline{S}}^{(\boldsymbol{r})}) - R_{l,f}(\boldsymbol{g}_{\overline{S}}^{(\boldsymbol{r})})}_{(a2)} + \underbrace{R_{l,f}(\boldsymbol{g}_{\overline{S}}^{(\boldsymbol{r})}) - R_l(f)}_{(a3)}.$$

(A.28)

The terms (a1) and (a2) of Eq. (A.28) can be upper-bounded as follows:

$$\text{(a1)} \leq \max_{\boldsymbol{g} \in \mathcal{G}(\boldsymbol{r}, \overline{\boldsymbol{S}})} |R_{l,f}(\boldsymbol{g}) - \widehat{R}_{l,f}(\boldsymbol{g})|, \tag{A.29}$$

$$\text{(a2)} \leq \widehat{R}_{l,f}(\boldsymbol{g}_{\overline{S}}^{(\boldsymbol{r})}) - R_{l,f}(\boldsymbol{g}_{\overline{S}}^{(\boldsymbol{r})}) \leq \max_{\boldsymbol{g} \in \mathcal{G}(\boldsymbol{r}, \overline{\boldsymbol{S}})} |R_{l,f}(\boldsymbol{g}) - \widehat{R}_{l,f}(\boldsymbol{g})|. \tag{A.30}$$

The term (a3) of Eq. (A.28) can be upper-bounded using Lemma A.1 as follows:

$$\text{(a3)} \leq |R_{l,f}(\boldsymbol{g}_{\overline{S}}^{(\boldsymbol{r})}) - R_l(f)|$$
$$\leq \left\{ 2\sqrt{R_l(f)} + \left( 2U_l \sum_{j \in [F^{\mathrm{w}}]} R_{01,j}(g_{\overline{S},j}^{(r_j)}) \right)^{\frac{1}{2}} \right\} \left( 2U_l \sum_{j \in [F^{\mathrm{w}}]} R_{01,j}(g_{\overline{S},j}^{(r_j)}) \right)^{\frac{1}{2}}. \tag{A.31}$$

Therefore, by applying Eqs. (A.29), (A.30), and (A.31) to Eq. (A.28), we obtain:

$$R_{l,f}(\boldsymbol{g}_{f,\overline{S}}^{(\boldsymbol{r})}) - R_l(f) \leq 2 \max_{\boldsymbol{g} \in \mathcal{G}(\boldsymbol{r}, \overline{\boldsymbol{S}})} |R_{l,f}(\boldsymbol{g}) - \widehat{R}_{l,f}(\boldsymbol{g})|$$
$$+ \left\{ 2\sqrt{R_l(f)} + \left( 2U_l \sum_{j \in [F^{\mathrm{w}}]} R_{01,j}(g_{\overline{S},j}^{(r_j)}) \right)^{\frac{1}{2}} \right\} \left( 2U_l \sum_{j \in [F^{\mathrm{w}}]} R_{01,j}(g_{\overline{S},j}^{(r_j)}) \right)^{\frac{1}{2}}. \tag{A.32}$$

From the uniform law of large numbers (Mohri et al., 2018), for any $\delta \in (0,1)$, the following holds with a probability of at least $1 - \delta$:

$$\max_{\boldsymbol{g} \in \mathcal{G}(\boldsymbol{r}, \overline{\boldsymbol{S}})} |R_{l,f}(\boldsymbol{g}) - \widehat{R}_{l,f}(\boldsymbol{g})| \leq 2\mathfrak{R}_n^*(\widetilde{\mathcal{G}}_{l,f}(\boldsymbol{r}, \overline{\boldsymbol{S}})) + U_l \sqrt{\frac{\log(2/\delta)}{2n}}. \tag{A.33}$$

Furthermore, by applying Eq. (A.33) to Eq. (A.32), we obtain that, for any $\delta \in (0,1)$, with probability at least $1 - \delta$, Eq. (4.10) holds:

$$R_{l,f}(\boldsymbol{g}_{f,\overline{S}}^{(\boldsymbol{r})}) - R_l(f) \leq 4\mathfrak{R}_n^*(\widetilde{\mathcal{G}}_{l,f}(\boldsymbol{r}, \overline{\boldsymbol{S}})) + 2U_l \sqrt{\frac{\log(2/\delta)}{2n}}$$
$$+ \left\{ 2\sqrt{R_l(f)} + \left( 2U_l \sum_{j \in [F^{\mathrm{w}}]} R_{01,j}(g_{\overline{S},j}^{(r_j)}) \right)^{\frac{1}{2}} \right\} \left( 2U_l \sum_{j \in [F^{\mathrm{w}}]} R_{01,j}(g_{\overline{S},j}^{(r_j)}) \right)^{\frac{1}{2}}. \tag{A.34}$$

$\square$

## A.6. Proof of Theorem 4.5

*Proof of Theorem 4.5.* By assumption, there exist true deterministic functions $g_j^* : \mathcal{X}^{\mathrm{o}} \rightarrow \mathcal{X}_j^{\mathrm{w}}$ for any $j \in [F^{\mathrm{w}}]$, and $(g_1^*, \dots, g_{F^{\mathrm{w}}}^*) \in \mathcal{G}$. In this case, for any $\boldsymbol{r}$ and $\overline{S}$, it holds that $\boldsymbol{g}^* \in \mathcal{G}(\boldsymbol{r}, \overline{S})$. Therefore, the following holds:

$$R_{01,j}(g_{\mathcal{G}(\boldsymbol{r}, \overline{S}),j}) = 0, \ \ \forall j \in [F^{\mathrm{w}}]. \tag{A.35}$$

For any $j \in [F^{\mathrm{w}}]$, define $\mathcal{G}_j(r_j, \overline{S}_j)$ as the set of hypotheses that satisfy the following two conditions: (i) each element is a solution obtained by methods that are guaranteed to achieve consistency (Cour et al., 2011; Feng et al., 2020; Xu et al., 2021; Natarajan et al., 2013; Ishida et al., 2017; Yu et al., 2018), and (ii) its empirical risk is at most $r_j$. As $n$ increases and $r_j \rightarrow 0$, the assumptions on $\overline{R}_{l_j}$ and the theoretical guarantees of these consistent methods for $g_j$ imply the following:

$$R_{01,j}(g_j) \rightarrow 0, \ \ \forall g_j \in \mathcal{G}_j(r_j, \overline{S}_j), \ \ \forall j \in [F^{\mathrm{w}}]. \tag{A.36}$$

Additionally, by assumption, there exists a true deterministic function $f^* : \mathcal{X} \rightarrow \mathcal{Y}$ for label prediction, and $f^* \in \mathcal{F}$. Hence, the following holds:

$$R_l(f_{\mathcal{F}}) = 0. \tag{A.37}$$

Thus, under the conditions of Theorem 4.3, the following holds for $f_{\boldsymbol{g},\overline{S}}$ obtained through LAC-dWFL's step (ii):

$$n \to \infty, \ \ R_{l,\boldsymbol{g}}(f_{\boldsymbol{g},\overline{S}}) \to 0. \tag{A.38}$$

Furthermore, using Theorem 3.1, and additionally letting $r_j \to 0$ as $n$ increases for any $j \in [F^{\mathrm{w}}]$, the following holds:

$$n \to \infty, \ \ R_l(f_{\boldsymbol{g},\overline{S}}) \to 0, \ \ \forall \boldsymbol{g} \in \mathcal{G}(\boldsymbol{r},\overline{S}). \tag{A.39}$$

Since $l$ is $L_l$-Lipschitz continuous and $f_{\boldsymbol{g},\overline{S}}$ is $L_f$-Lipschitz continuous, using Talagrand's lemma (Shalev-Shwartz & Ben-David, 2014), the following holds:

$$\mathfrak{R}_n^*(\tilde{\mathcal{G}}_{l,f}(\boldsymbol{r},\boldsymbol{S})) \leq L_l L_f \mathfrak{R}_n^*(\mathcal{G}(\boldsymbol{r},\boldsymbol{S})). \tag{A.40}$$

Consequently, if $\mathfrak{R}_n^*(\mathcal{G}(\boldsymbol{r},\boldsymbol{S}))$ and $\mathfrak{R}_n^*(\mathcal{G}_j(r_j,\overline{S}_j))$ for any $j \in [F^{\mathrm{w}}]$ are monotonically decreasing with respect to $n$ and converge to 0, the error bound established in Theorem 4.4 converges to 0 as $n \to \infty$.

$\square$

# B. Detail Information of Experiments

## B.1. Details of Datasets

We used four real-world datasets: *Adult* (Becker & Kohavi, 1996), *Bank Marketing* (Moro & Cortez, 2014), *kick* (Vanschoren et al., 2013), and *Census-Income (KDD)* (cen, 2000; Dua & Graff, 2017) in Section 5. In Section C.1, we additionally used two datasets: *Default of Credit Card Clients* (Yeh, 2009) and *Diabetes 130-US Hospitals for Years 1999–2008* (Kahn). We will refer to them as *Adult*, *Bank*, *Kick*, *Census*, *Default*, and *Diabetes*, respectively. These datasets can be downloaded from UCI Machine Learning Repository (Dua & Graff, 2017) or OpenML (Vanschoren et al., 2013). Table B.1 summarizes the characteristics of these datasets. All binary features were set to take values of either 0 or 1, all categorical features were encoded using one-hot encoding, and all continuous features were scaled to fall within the range $[0, 1]$. For the Adult, Bank, Kick, Default, and Diabetes datasets, all available samples were used. For the Census dataset, experiments were conducted using 50,000 randomly sampled data points.

*Table B.1.* Outline of datasets. binary, categorical, and numerical represent the number of features of each type, respectively.

| dataset | Adult | Bank | Kick | Census | Default | Diabetes |
|---|---|---|---|---|---|---|
| data size | 48842 | 45211 | 72983 | 299285 | 30000 | 101766 |
| binary | 1 | 3 | 3 | 3 | 1 | 10 |
| categorical | 7 | 5 | 12 | 22 | 2 | 12 |
| numerical | 6 | 8 | 16 | 10 | 20 | 10 |
| target | binary | binary | binary | binary | binary | 3 classes |

## B.2. Details of Experimental setup

We summarize the settings of feature estimation models $\boldsymbol{g}$ and label prediction model $f$. Both $\boldsymbol{g}$ and $f$ were implemented using two-layer perceptrons with hidden layers of width 500 and ReLU as an activation function. The optimization algorithm used across all models is Adam (Kingma & Ba, 2014), with the following hyperparameters: a learning rate of 0.0005, batch size of 512, 100 epochs, and a weight decay of 0.0002. Logistic loss was used for training $f$.

We summarize the method for calculating the error bound presented in Theorem 4.2 for this experiment. The parameter $\delta$ was set to 0.01. Additionally, assuming that $\mathcal{F}$ is sufficiently expressive, we set $R_l(f_{\mathcal{F}}) = 0$. During inference, the predicted label from $f$ was determined based on the largest output value of $f$. Consequently, scaling the outputs of $f$ does not affect the inference results. Thus, we assumed that the maximum value of each element in $f$'s output is 1 and set $U_l = 2.0$. Since the loss function $l$ for label prediction is the logistic loss, which is 1-Lipschitz continuous, it follows that $\mathfrak{R}_n^*(\tilde{\mathcal{F}}_l) = \mathfrak{R}_n^*(\mathcal{F})$ and $\mathfrak{R}_n^g(\tilde{\mathcal{F}}_l) = \mathfrak{R}_n^g(\mathcal{F})$ (Lemma 26.9 in (Shalev-Shwartz & Ben-David, 2014)). The terms $\mathfrak{R}_n^*(\mathcal{F})$ and $\mathfrak{R}_n^g(\mathcal{F})$ were computed using the upper bound on the Rademacher complexity of a multilayer perceptron derived by Neyshabur et al. (Theorem 1 in (Neyshabur et al., 2015)). To compute this upper bound, it is necessary to determine $\mu$, which represents the upper bound on the $l_p$-norm of all parameters of $f$, as well as the value of $p$. In this experiment, $p = 2$ was chosen. Furthermore, since it is possible to scale all parameter values of $f$ without affecting the inference results for predicting a single label, we set $\mu = 1$.

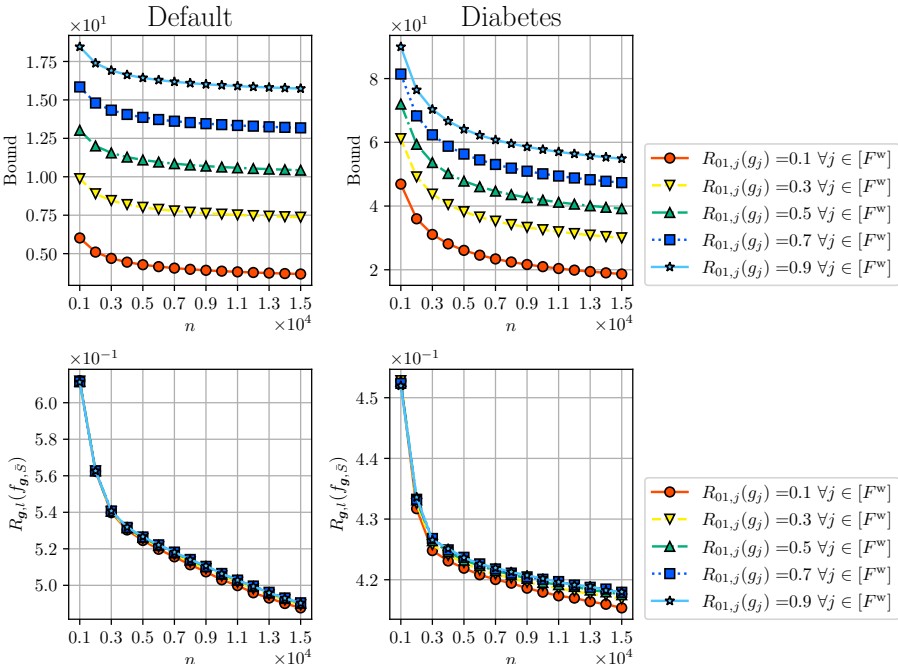

*Figure C.3.* A comparison between $R_{l,\boldsymbol{g}}(f_{\boldsymbol{g},\overline{S}})$ and the error bound derived in Theorem 4.2, for various estimation errors of $\boldsymbol{g}$ using Default and Diabetes datasets.

# C. Additional Experiments

## C.1. Additional Datasets

In Section 5, we validated our theoretical results through numerical experiments using real-world datasets. In this section, we further examine the validity of our findings on two additional datasets that were not used in Section 5.2. The datasets used here are Default (Yeh, 2009) and Diabetes (Kahn), with details provided in Appendix B.1. Figure C.3 presents the results of the same experimental procedure applied to these datasets, following the methodology outlined in Section 5.2. In these datasets, variations in the estimation accuracy of the exact values of WFs resulted in only minor changes in the risk of the downstream task. As a consequence, visualizations similar to those in Figure 5.2 significantly compromised readability. To address this, we separately plot the observed risks and the theoretical bound derived from Theorem 4.7.

From Figure C.3, we observe that a smaller value of $R_{01,j}(g_j)$ leads to a greater reduction in $R_{\boldsymbol{g},l}(f_{\boldsymbol{g},\overline{S}})$ as $n$ increases. This trend is consistent with the behavior of the error bounds illustrated in the same figure. Regarding the rate of decrease in both $R_{\boldsymbol{g},l}(f_{\boldsymbol{g},\overline{S}})$ and the bound with respect to $n$, we find that their sensitivity to changes in $R_{01,j}(g_j)$ is less pronounced in the Default dataset than in the Diabetes dataset. This difference can be attributed to the fact that the Default dataset contains only two WFs, whereas the Diabetes dataset includes four. Consequently, the influence of WFs on downstream tasks is inherently smaller in the Default dataset. Therefore, from the results on these two additional datasets, we further confirm that our derived error bound in Theorem 4.7 successfully capture the relationship between the rate of decrease in $R_{\boldsymbol{g},l}(f_{\boldsymbol{g},\overline{S}})$ with increasing $n$ and the value of $R_{01,j}(g_j)$.

## C.2. Comparison with the Case WFs are Used for the Inputs of $f$

In this section, we compared the risk of the $f$ obtained by directly using $\overline{\boldsymbol{X}}^{\mathrm{w}}$ for training (i.e., $\boldsymbol{g}(\boldsymbol{X}^{\mathrm{o}}) = \overline{\boldsymbol{X}}^{\mathrm{w}}$) with the experimental results on risks presented in Section 5. We focused on the four datasets used in Section 5, as the two additional datasets examined in Section C.1 exhibited a relatively weaker dependency of the risk of $f$ on the estimation accuracy of $\boldsymbol{g}$. The training procedure for $f$ remained identical to that described in Section 5. By conducting this comparison, we aim to provide empirical insight into how accurately $\boldsymbol{g}$ must estimate the exact values of WFs in order to improve the generalization performance of $f$.

Figure C.4 shows the results under the setting where all WFs are complementary features (CFs). Each observation of a

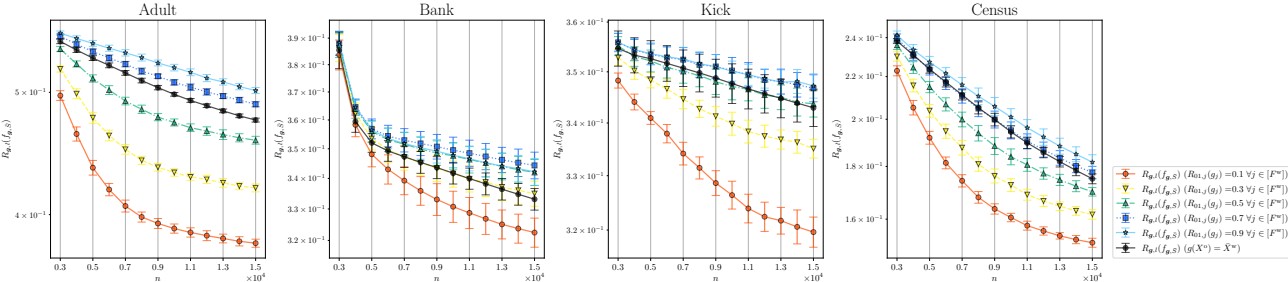

*Figure C.4.* Comparison between the risks in Section 5 and the risk of $f$ when $g(X^\circ) = \overline{X}^w$. The results correspond to the case where all WFs are CFs, and both the mean and standard deviation over five trials are reported.

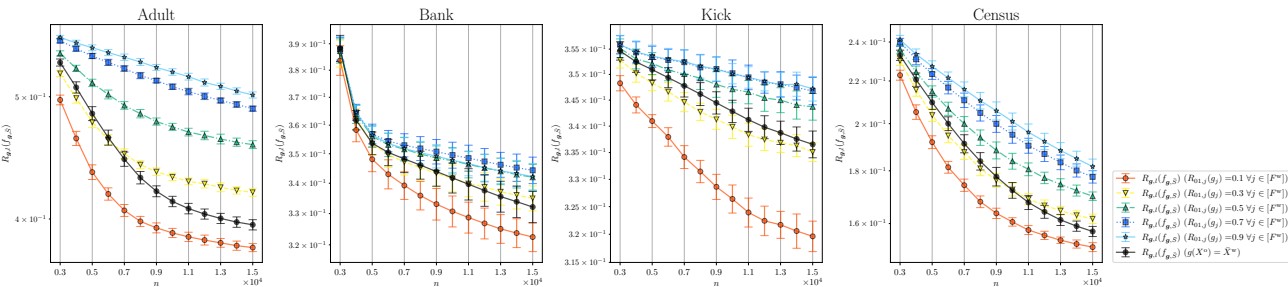

*Figure C.5.* Comparison between the risks in Section 5 and the risk of $f$ when $g(X^\circ) = \overline{X}^w$. The results correspond to the case where each WF is observed as a set of size two including the exact value, with mean and standard deviation over five runs reported.

CF is sampled uniformly from all values except the exact one. From this figure, we observe that for the Adult, Kick, and Census datasets, a classification error of $g_j$ below $0.5$ is sufficient to achieve a model $f$ that outperforms the baseline where $g(X^\circ) = \overline{X}^w$. In contrast, for the Bank dataset, the classification error of $g_j$ must be below at least $0.3$ to achieve similar improvement.

Figure C.5 presents the results for the setting in which each observed value of every WF is represented as a set of size two that includes the exact value. The additional value in the set, apart from the exact value, is uniformly sampled at random. When using such $\overline{X}^w$ as input to $f$, each WF is encoded as a one-hot vector, where the entries corresponding to the values in the observed set are set to one. These vectors are then used as inputs to $f$. From Figure C.5, in contrast to the case where WFs are CFs, we observe that $g$ yielding better performance than the case where $g(X^\circ) = \overline{X}^w$ depends on $n$. This difference is attributed to the fact that $\overline{X}^w$ always includes the exact value, whereas estimation via $g$ inevitably produces instances with incorrect values due to estimation errors. These results suggest that, for estimating the exact values of WFs, it may be more effective to output a probability distribution over possible values, rather than predicting a single deterministic value. Accordingly, developing methods where $g$ produces a distribution as output, along with establishing a theoretical framework that accommodates such $g$, are important future directions for WFL. The findings in this paper are considered to provide a fundamental basis for such extensions.

## D. Limitation

In this paper, we consider the basic framework of WFL, where a feature estimation model is constructed for each WF independently, using OFs as inputs. Accordingly, our framework does not consider approaches that incorporate dependencies among WFs into the construction of feature estimation models. For example, we do not address methods that first estimate the exact values of a given WF and then use those estimates as input features to infer the exact values of other WFs, or approaches that jointly estimate the exact values of multiple WFs within a single model. Developing such approaches is of practical importance and is left for future work. Furthermore, analyzing such methods would require a theoretical framework for quantifying how well the dependencies among WFs are captured, as well as for modeling these dependencies.

Theoretical insights presented in this paper are considered to provide a foundational basis for such future investigations.

For similar reasons, methods that construct $f$ and $g$ as a single unified model and train it jointly are not yet covered by the analysis presented in this paper. Developing and analyzing such methods also constitutes an important direction for future research.

Our analysis does not account for the feature importance of each WF in the downstream prediction task. Intuitively, the estimation accuracy for WFs that are more strongly related to the downstream target variable should have a greater impact on the error bound for learning $f$, compared to those WFs that are weakly related or irrelevant. However, the error bound we derived does not account for such feature importance and thus cannot differentiate the relative contributions of individual WFs. Therefore, deriving an error bound that incorporates feature importance remains an important direction for future work. It is considerd that the results presented in this paper provide a solid foundation for such an extension.

