# OpenReview forum: "A Unified Framework for Generalization Error Analysis of Learning with Arbitrary Discrete Weak Features"
_ICML.cc/2025/Conference — ICML 2025 poster_

### Official Review · Reviewer_VwNg · 2025-02-20

**Overall Recommendation:** 3

**Summary:**

The paper presents a unified formalization and theoretical analysis of discrete Weak Features Learning (WFL) to handle learning with arbitrary discrete weak features (WFs). It introduces a set of algorithms that jointly learn the estimation model for WFs and the predictive model for a downstream task while conducting a generalization error analysis under finite-sample conditions. Additionally, the authors provide empirical results that support their findings.

**Claims And Evidence:**

The authors offered both theoretical and empirical analyses to support all of their claims.

**Essential References Not Discussed:**

The literature review is extensive.

**Experimental Designs Or Analyses:**

The experimental design effectively supports the theoretical result. However, the absence of a baseline method comparison means there is no performance evaluation against existing methods. This limitation is unfavorable for practitioners.

**Methods And Evaluation Criteria:**

1. The proposed methods and evaluation criteria are suitable for the problem.

2. However, no benchmark methods are provided to demonstrate and compare the performance of the proposed methods.

**Other Comments Or Suggestions:**

1.Suggestion: Add an algorithm table to summarize the methods.

**Other Strengths And Weaknesses:**

**Other Strengths:**

1. The paper's motivation and structure are clear and reader-friendly.

2. The notation and setup are also clearly presented.

3. Codes are provided in the supplementary materials, which benefit practitioners.

**Other Weaknesses:**

1.Theorem 4.3 and Theorem 4.5 lack theoretical contributions.

1.1 These two results provide only consistency findings; no convergence rates are presented.

1.2 The proofs of these theorems rely predominantly on results from existing references.

**Questions For Authors:**

1. Is it possible to add the details of proof for my listed confusions in the Theoretical Claims* section?

**Relation To Broader Scientific Literature:**

This study offers comprehensive theoretical insights into various problem settings, such as ItR and CFL, involving discrete WFs. This is beneficial to the statistical learning community.

**Theoretical Claims:**

I checked the proof of the theoretical claims. Most of them are correct except that some key steps are not clear and demonstrating enough: for example

1.P11. A.1 The second inequality arises from the decomposition of the 0-1 loss. This step is not clear enough.

2.A.2 How (a1) and (a2) leads to (A.12) is not clear enough.

3.A.3 P15, top two inequalities under From the uniform law of large numbers... (line 827-829) ? Why do they hold? Here is not clear enough.

4.A.5 line 871, why does the second equality hold?

5.A.5 line 876-886, how the analysis above this inequality leads to this result need to analyze clearly. This is not a simple step.

---

> ### Author Rebuttal · Authors · 2025-04-01
>
> Thank you for your careful evaluation and valuable suggestions to improve our paper.
>
> **Comparison with Benchmark Methods:**
>
> The most relevant benchmark is using WFs directly as input features, because our framework is designed to improve upon this approach by accommodating various learning methods for learning $f$ and $g$.
>
>
> Since this study focuses on the theoretical understanding of WFL, our experiments were conducted to validate the theoretical results rather than compare against the benchmark method. However, we recognize the importance of empirical comparisons with the benchmark method and will include them in the Appendix of the Camera Ready version.
>
> We are currently conducting additional experiments, and preliminary results indicate that when $g$ is randomized, our framework outperforms the baseline if $g$’s error rate is below 0.5. Thus, for problem settings where the exact values of WFs can be well estimated, our framework could be a valuable tool for practitioners.
>
> **Clarification on the Second Inequality in A.1:**
>
> In p.11 lines 567–568, the 0-1 loss is 0 when $g$ correctly predicts the exact values of all WFs and 1 otherwise. In contrast, in lines 570–571, the sum of the 0-1 losses is 0 when $g$ correctly predicts all WFs and equals $k \ge 1$ when $g$ mispredicts $k$ WFs. Based on this fact, the second inequality in p.11 Eq.(A.1) holds.
>
> **Derivation from (a1) and (a2) in Eq.(A.11) to Eq.(A.12) in A.2:**
>
> Eq. (A.11) shows that the LHS of Eq. (A.12) is upper-bounded by the expectation of the sum of (a1) and (a2). After Eq.(A.11), we analyze (a1) and (a2) separately. Substituting these results into the RHS of Eq.(A.11) and simplifying the expectation operation yields the RHS of Eq.(A.12), confirming its validity.
>
> **Clarification on the Two Inequalities in A.3 (P.15, Line 770):**
>
> The question refers to lines. 827–829, but the uniform law of large numbers is actually used from l. 770 onward, so we interpret the comment as referring to that part. This law ensures uniform convergence of the empirical expectation to the true expectation over the function class, given that the finite samples are i.i.d. In the top two equations on p.15, the samples used for each empirical risk are indeed i.i.d., validating these inequalities. We will add a reference to Chapter 3.1 (pp. 30–34) in [Mohri 2018] in the Camera Ready version for further clarification.
>
> **Justification of the Second Equality in A.5 (Line 871):**
>
> Thank you for your valuable comment. We acknowledge that the second equality in line 871 does not hold due to a minor mistake. We will not use this equality and instead consider only the first inequality in line 871. Additionally, we will replace $\mathfrak{R}^*\_{n}(\mathcal{G}\_j(r\_j, \bar{S}\_j))$ with $\mathfrak{R}^*\_{n}( \tilde{\mathcal{G}}\_j(r\_j, \bar{S}\_j))$ in lines 874–887 and in Eq. (4.10) of Theorem 4.4. We explain why this modification is sufficient in the next section.
>
> This correction does not affect the subsequent discussion and the theoretical results of the paper. This is because, in the discussion from Theorem 4.4 onward, including Theorem 4.5, the original assumption on the convergence rate of $\mathfrak{R}^*\_{n}(\mathcal{G}\_j(r\_j, \bar{S}\_j))$  is simply replaced with the same assumption for $\mathfrak{R}^*\_{n}( \tilde{\mathcal{G}}\_j(r\_j, \bar{S}\_j))$, and these assumptions are equally valid. We will correct this in the camera-ready version.
>
> **Clarification on the Derivation of A.5 (Lines 876-886):**
>
> The equations in A.5 (lines. 876–886) can be derived by applying Eq. (A.33) and the equation in lines. 870–873 to the RHS of Eq. (A.32). We acknowledge that this step was not explicitly stated and will clarify it in the Camera Ready version.
>
> **Theoretical Contributions of Theorems 4.3 and 4.5:**
>
> For Weakness 1.1, the discussion on convergence rates appears follows Theorems 4.2 and 4.4, where error bounds are established (lines 258–320, 356–378). We used “rate of decrease” in the discussion but recognize that “convergence rate” is more precise. We will revise this in the Camera Ready version.
>
> Regarding Weakness 1.2, our study provides insights that go beyond a simple combination of existing theoretical results on $f$ and $g$. While their learning methods rely on prior approaches, a theoretical analysis of WFL requires explicitly modeling their interactions. We address this through Theorems 4.2 and 4.4. Theorems 4.3 and 4.5 establish the consistency of WFL by integrating existing theories using the results of Theorems 4.2 and 4.4. We acknowledge that this was not sufficiently clear and will provide further clarification in the Camera Ready version.
>
> **Procedure of Our Algorithm Class:**
>
> We will make the algorithm class procedure easier to understand in the Camera Ready version.
>
> Reference:
>
> [Mohri 2018] Mohri, M. Foundations of machine learning, 2nd edition. 2018.

---

> > ### Comment · Reviewer_VwNg · 2025-04-02
> >
> > Thank you for your thorough response to my reviews. All my concerns have been addressed. I will maintain my overall ratings, and I'm happy to see that some of my suggestions may be included in the next version.
> >
> > Clarification on the Two Inequalities in A.3 (P.15, Line 770):
> >
> > My original question is about the issue you interpreted in the rebuttal, which was caused by my typo of line numbers.

---

> > > ### Author Response · Authors · 2025-04-02
> > >
> > > Thank you again for your valuable time and dedicated effort. It was gratifying to know that our response addressed your concerns. We especially appreciate the insightful comments and constructive feedback you shared throughout the review process.

---

### Official Review · Reviewer_Wr5c · 2025-03-09

**Overall Recommendation:** 3

**Summary:**

In this paper, the authors propose a unified framework called weak feature learning which accommodates arbitrary discrete weak features and a broad range of learning algorithms. The authors also introduce a class of algorithms that learn both the estimation model for weak features and the predictive model for downstream task and present generalization error analysis, as well as, theoretical conditions necessary for achieving consistency of the learning method.

 ### update after rebuttal

The authors have answered my questions and I will maintain my score.

**Claims And Evidence:**

Yes.

**Essential References Not Discussed:**

Yes.

**Experimental Designs Or Analyses:**

Yes.

**Methods And Evaluation Criteria:**

Yes.

**Other Comments Or Suggestions:**

See above.

**Other Strengths And Weaknesses:**

1. Once Lemma 4.1 is established, why Theorem 3.1 is needed?

2. Since the features $x_w$ and $x_0$ are disjoint set of features, when is it possible to learn weak features $x^w$ from normal features $x_0$? Is it inherently assumed that $x_0$ contains enough  information regrading $x_w$ so that the estimation function $g$ makes sense?

3. How would one design experiment to demonstrate the effectiveness of Theorem 4.4?

**Questions For Authors:**

See above.

**Relation To Broader Scientific Literature:**

The results of this paper elucidates the interplay between estimation error of weak features and prediction error of a downstream task which seems to be an important problem.

**Theoretical Claims:**

I did not check all the proofs. However, in the proof of Theorem 3.1, after the second inequality, $M_l$ should be $U_l$.

---

> ### Author Rebuttal · Authors · 2025-04-01
>
> Thank you for your thoughtful and thorough review of our manuscript, and for pointing out areas for further clarification and improvement.
>
> **Necessity of Theorem 3.1 Given Lemma 4.1 (about Question 1.):**
>
> We agree that Lemma 4.1 provides a tighter bound and that Theorem 3.1 can be derived from it. However, we introduced Theorem 3.1 to improve the clarity of our discussion. While Lemma 4.1 focuses on expressing the relationship between the risks of $f$ and $g$ for detailed analysis, Theorem 3.1 is intended to establish the validity of our proposed framework. Specifically, Theorem 3.1 explicitly illustrates the correspondence between the risk $R_l(f)$ in standard supervised learning and the objective function of WFL $R^{\mathrm{dWFL}}_{l, \lambda}(f)$. This correspondence is not straightforward to interpret directly from the inequality in Lemma 4.1. Since this clarification was not explicitly stated in our paper, we will include it in the Camera Ready version.
>
> **Dependence on Informative Ordinary (Normal) Features (about Question 2.):**
>
> Your understanding is correct. Since our framework is based on constructing a model that estimates $X^{\mathrm{w}}$ from $X^{\mathrm{o}}$​, accurate estimation of $X^{\mathrm{w}}$ is feasible when $X^{\mathrm{o}}$ is sufficiently correlated with $X^{\mathrm{w}}$. However, while a strong correlation between $X^{\mathrm{o}}$​ and $X^{\mathrm{w}}$​ is a practical condition for achieving good performance, our theoretical results hold regardless of this assumption.
>
> **Experimental Design to Demonstrate the Effectiveness of Theorem 4.4 (about Question 3.):**
>
> To demonstrate the effectiveness of Theorem 4.4, we would need to generate multiple functions $f$ with varying accuracy and examine whether the learned results of $g$ exhibit a trend similar to the error bound provided by Theorem 4.4.
> There are two primary methods to construct $f$ with varying accuracy: (1) training $f$ directly using different learning settings and (2) constructing $f$ in a manner similar to $g$ in Section 5.2.
>
> The first approach can be implemented by training a neural network while carefully adjusting training hyperparameters and dataset composition. However, ensuring that these adjustments are not arbitrary is challenging, which may introduce biases and compromise the justification and validity of the experimental results.
>
> The second approach, following Section 5.2, is not feasible because the constructed $f$ would make $g$ untrainable. Since $f$ makes random predictions with a fixed accuracy, the empirical risk $\hat{R}_{l,f}(g)$ becomes non-differentiable with respect to $g$, preventing effective learning.
>
> Due to these limitations, this experimental design is challenging. However, we note that the error bound for $f$ in Theorem 4.2 shares a highly similar form with the bound for $g$ in Theorem 4.4. Given the verification of $f$'s bound in Section 5.2, we can reasonably infer the validity of $g$'s bound as well.
>
>
> **Typographical Error in Theorem 3.1 Proof:**
>
> Thank you for pointing this out. We confirm the mistake and will correct it in the Camera Ready version.

---

> > ### Comment · Reviewer_Wr5c · 2025-04-03
> >
> > Thank you for your reply. Most of my questions have been ansered. I will maintain my score.

---

> > > ### Author Response · Authors · 2025-04-05
> > >
> > > Thank you again for your time and thoughtful review. We’re pleased to hear that our responses have clarified your concerns.

---

### Official Review · Reviewer_udjK · 2025-03-09

**Overall Recommendation:** 3

**Summary:**

The paper presents a unified framework called WFL for analyzing generalization error in learning tasks involving arbitrary discrete WFs. The authors propose a risk-based formulation that accommodates various types of WFs and a broad range of learning algorithms.

## update after rebuttal

The author's response has addressed some of my concerns, and I will keep the score.

**Claims And Evidence:**

The claims made in the paper are generally supported by theoretical analysis and experimental validation. However, the experimental results are limited to a few datasets, and the impact of different types of WFs on the learning process could be further explored.

**Essential References Not Discussed:**

I am not an expert in this specific area, so I have no suggestion on more refrerences.

**Experimental Designs Or Analyses:**

The experimental design is sound, and the authors provide a clear explanation of their setup and methodology.

**Methods And Evaluation Criteria:**

The proposed methods and evaluation criteria are appropriate for the problem at hand. The authors use a combination of theoretical analysis and empirical validation to demonstrate the effectiveness of their framework.

**Other Comments Or Suggestions:**

- The paper is well-written, but there are a few minor grammatical errors and typos that could be corrected. For example, in Section 3.2, the phrase "The primary factor reducing explainability is the inaccuracy of information provided by WFs" could be rephrased for clarity.

- The notation in some of the equations could be more consistent. For example, in Equation (3.4), the use of $\lambda$ as a weighting parameter is clear, but the notation could be more consistent with the rest of the paper.


***Note:*** the template of this paper seems to be different from the official template.

**Other Strengths And Weaknesses:**

The paper's main strengths lie in its theoretical contributions and the unified framework it provides for analyzing generalization error in learning tasks with weak features.  However, the impact of different types of WFs on the learning process could be further explored.

**Questions For Authors:**

- Can the authors provide more details on the computational complexity and scalability of the proposed methods, particularly for large-scale datasets?

- How does the proposed framework handle continuous weak features, and are there any limitations in this regard?

- Could the authors discuss any potential challenges or limitations in applying the proposed framework to real-world applications, particularly in domains with complex data distributions?

**Relation To Broader Scientific Literature:**

The paper builds on existing work in WSL and extends it to the problem of learning with WFs. The authors discuss related work in WSL, including semi-supervised learning and learning with label noise, and provide a unified framework that accommodates various forms of WFs. The paper also connects to prior work on ItR and CFL, providing a theoretical foundation for these methods.

**Theoretical Claims:**

I have reviewed the theoretical claims and proofs presented in the paper. The proofs appear to be correct, and the authors provide detailed derivations for their error bounds and consistency conditions. However, I am not an expert in this specific area, so I cannot fully verify the correctness of all the theoretical results.

---

> ### Author Rebuttal · Authors · 2025-04-01
>
> We sincerely appreciate the time and effort the reviewer dedicated to evaluating our work and are grateful for your insightful feedback and constructive suggestions.
>
> **The Impact of Different Types of WFs on the Learning Process and Experimental Datasets:**
>
> Our framework and analysis hold regardless of the type of discrete WFs, which you have also acknowledged in "Relation to Broader Scientific Literature." This generality is possible because, within our framework, the learning of $f$ and $g$ is decoupled: differences in WF types affect only the learning of $g$, while during the learning of $f$, the WF types are concealed, and only $g$ influences $f$. Also, the effect of WF types on $g$ can be thoroughly investigated through existing theories, such as those related to PLL and CLL. Therefore, to validate our theoretical results, it suffices to examine the behavior of $f$ under different levels of $g$'s accuracy rather than testing specific WF types.
>
> Meanwhile, we acknowledge your concern regarding the limited variety of datasets used in our experiments. We take your feedback seriously and are conducting additional experiments on a medical dataset and a dataset related to the prediction of defaults. The results will be included in the Appendix of the Camera Ready version. While our additional experiments are still ongoing, preliminary results indicate qualitatively consistent findings across new datasets.
>
>
> **Computational Complexity and Scalability, and Limitations in Real-World Applications (about Q.1 & .3) :**
>
> We agree that computational complexity, scalability, and handling complex data distributions are crucial considerations for real-world applications.
>
> Our proposed framework is designed to integrate existing learning algorithms for estimating WFs and predicting a downstream task. Consequently, the computational complexity, scalability, and adaptability to complex data distributions depend on the choice of these algorithms. As for computational complexity, in sequential learning, the computational complexity is determined solely by the sum of computational complexities of the methods applied to learn $f$ and $g$, respectively. In iterative learning, the only additional factor is the number of iterations. Therefore, by selecting appropriate methods within our framework, we can ensure scalability and adaptability to complex data distributions while controlling computational complexity. This flexibility is largely due to the simplicity of our framework.
> However, this simplicity also imposes certain limitations, particularly in leveraging domain knowledge for real-world applications. For instance, our framework does not support approaches where dependencies among WFs are exploited to estimate certain WFs first and use their results for estimating other WFs. Addressing this limitation is an important direction for extending our framework, and we plan to explore this in future research.
>
> Since these points were not explicitly discussed in our paper, we appreciate your valuable feedback and will include additional clarifications in the Camera Ready version.
>
>
> **Handling Continuous Weak Features (about Q.2) :**
>
> We have addressed a similar question from Reviewer guZX. Our framework can be applied to continuous WFs in almost the same manner as for discrete WFs. Please refer to our response (the response “Extending the Framework to Handle Continuous Weak Features” for reviewer guZX) there for details.
>
> **Grammatical Errors and Typos:**
>
> Due to time constraints, we are unable to make extensive revisions at this stage. However, we plan to have the paper professionally proofread before the Camera Ready submission.
>
>
> **Notation Consistency and Template Confirmation:**
>
> Thank you for your feedback. We will carefully review the notation to ensure consistency throughout the paper. Regarding the template, we used the official ICML 2025 template downloaded from the conference website, but we will verify it again for correctness.

---

### Official Review · Reviewer_guZX · 2025-03-13

**Overall Recommendation:** 3

**Summary:**

This paper introduces a unified framework for Weak Feature Learning (WFL), which aims to address the challenge of learning with arbitrary discrete weak features (WFs)—features that are incomplete, erroneous, or ambiguous due to various real-world constraints. The authors propose a risk-based formulation that jointly optimizes a feature estimation model to approximate the true WFs and a label prediction model to perform the downstream task. Theoretical guarantees on consistency and error propagation are validated empirically across real-world datasets.

**Claims And Evidence:**

The claims are well-supported by both theoretical analysis and empirical evidence.

**Essential References Not Discussed:**

The paper adequately cites major prior works.

**Experimental Designs Or Analyses:**

The experiments are well-designed and convincingly validate the theoretical framework.

**Methods And Evaluation Criteria:**

The evaluation strategy is well-structured and rigorously tests the framework’s effectiveness.

**Other Comments Or Suggestions:**

Investigate merging causal inference to distinguish feature ambiguity from noise.

**Other Strengths And Weaknesses:**

Strengths
1. This is the first work to unify WFL methods with finite-sample error bounds and consistency proofs.
2. Validated on real-world datasets; code and appendix enhance reproducibility.
3. Bridges ItR, CFL, and weakly supervised learning under a single framework.
Weaknesses
1. The proposed framework excludes continuous weak features, restricting applicability.
2. Rademacher complexity computations may not scale to high-dimensional data.

**Questions For Authors:**

1. How would you extend the framework to handle continuous weak features (e.g., sensor readings with Gaussian noise)?
2. How does the framework perform under adversarial weak features (e.g., manipulated inputs)?

**Relation To Broader Scientific Literature:**

1 . Weakly supervised learning (WSL) (e.g., complementary labels, positive-unlabeled learning).
2. Imputation-based methods (ItR) and feature refinement strategies (CFL).

**Theoretical Claims:**

These theoretical results are well-supported and logically sound.

---

> ### Author Rebuttal · Authors · 2025-04-01
>
> Thank you for your thoughtful review and constructive feedback on our work.
>
> **Extending the Framework to Handle Continuous Weak Features:**
>
> Extending weak features learning (WFL) to continuous weak features (WFs) is an important research direction. We have investigated this and found that replacing the 0-1 loss-based risk in discrete WFL with a mean squared error (MSE)-based risk yields a parallel theoretical result. However, deriving this required a completely different approach, making it difficult to unify both theories in a single paper. Thus, we have compiled continuous WFL separately.
>
> Key differences in continuous WFL are as follows:
> * Continuous WFL requires handling WFs with continuous-valued uncertainties, such as missing or misobserved values, added Gaussian noise, or intervals containing the exact value.
>
> * The estimation error of $g$ in continuous WFL must be measured using metrics suitable for continuous values, such as MSE.
>
> * The theoretical framework of discrete WFL is based on the 0-1 loss, and its derivation heavily relies on the discreteness of WFs (e.g., Theorem 3.1 and Lemma 4.1). Consequently, this theory cannot be directly applied to continuous WFL.
>
> To establish a parallel theory, we derive an inequality that upper-bounds Lemma 4.1’s LHS by the product of the generalization error of $f$ and the MSE of $g$. Lemma 4.1’s LHS can be bounded using total variation distance and further upper-bounded by KL divergence via Pinsker’s inequality. Assuming $g$’s prediction follows a gaussian distribution, its log-likelihood corresponds to its MSE, allowing the desired inequality to be derived.
>
> This result establishes a continuous WFL theory aligned with discrete WFL, suggesting that WFL provides a unified framework for both discrete and continuous WFs.
>
> **Scalability of Rademacher Complexity Computations:**
>
> We agree that computing Rademacher complexity is challenging for high-dimensional data. However, this is a general issue for all analyses relying on Rademacher complexity, not a specific limitation of our work.
>
> Our results provide valuable theoretical insights, such as the convergence rates of error bounds for $f$ and $g$, their mutual interactions, and the impact of WF properties on $f$'s learning. These contributions remain valid regardless of whether Rademacher complexity can be explicitly computed.
>
> If one needs to estimate error bounds in practice, approximation methods can facilitate Rademacher complexity computation. For instance, dimensionality reduction techniques can compress high-dimensional data before computing Rademacher complexity.
>
> Importantly, the computability of Rademacher complexity does not affect the validity of our theoretical analysis, which remains robust even in high-dimensional settings.
>
> **Incorporating Causal Inference into WFL:**
>
> We have not yet explored integrating causal inference into WFL, but we agree that this extension is highly intriguing. Inspired by your suggestion, we considered an approach of modeling the observational process of WFs using a Structural Causal Model (SCM).
>
> The ambiguity in WFs' observations arises from factors such as observational noise, anonymization processes, or the probabilistic nature of their exact values. By modeling the generation process of WFs with an SCM, we believe it is possible to separate these effects, leading to improved estimation accuracy of WFs' exact values and a better understanding of the factors of ambiguity. A potential approach involves Bayesian estimation of an SCM in which the parameters related to observational noise, anonymization, WFs' exact values, and their probabilistic components are treated as latent variables.
>
> From a theoretical perspective, an important question is how the structure of the SCM and the quality of latent variable estimation influence the error bounds of $f$. If such error bounds can be derived, they could provide insights into the theoretical foundations of model selection criteria.
>
>
> **Performance Under Adversarial Weak Features:**
>
> While our framework was not initially designed for adversarial WFs, your comment prompted us to examine its relevance to such scenarios. We identified two types of adversarial manipulation and found that our framework effectively captures their impact on learning behavior.
>
> First, adversarial perturbations to observed values of WFs in training data would likely degrade the accuracy of $g$. Our error bounds account for this, reflecting the increased difficulty of the downstream task due to reduced $g$’s accuracy.
>
> Second, if only the most informative features are designated as WFs, accurately estimating their values becomes critical. As discussed in Section 4.2, our framework expresses the increased learning difficulty when these exact values cannot be precisely estimated.

---

### Decision · Program_Chairs · 2025-05-01

**Decision:**

Accept (poster)

**Comment:**

This paper is concerned with learning with Weak Features. It provides a framework which consists of formulation of the problem of learning Discrete Weak Features and using this formulation several algorithms are derived. It provides proofs of consistency as well. These are novel contributions and should be interesting to ICML community.
However,  during the discussion phase there was no champion and thus it would be difficult to argue for a strong accept. Additional results involving Causality or continuous values.